# BUILDING A SUBSPACE OF POLICIES FOR SCALABLE CONTINUAL LEARNING

**Jean-Baptiste Gaya**
Meta AI Research
CNRS-ISIR, Sorbonne University, Paris, France
`jbgaya@meta.com`

**Thang Doan**
McGill University, Mila
(Now at Bosch Research)
`thang.doan@mail.mcgill.ca`

**Lucas Caccia**
McGill University, Mila
`lucas.page-caccia@mail.mcgill.ca`

**Laure Soulier**
CNRS-ISIR, Sorbonne University, Paris, France
`laure.soulier@isir.upmc.fr`

**Ludovic Denoyer**
Ubisoft France
`ludovic.denoyer@ubisoft.com`

**Roberta Raileanu**
Meta AI Research
`raileanu@meta.com`

## ABSTRACT

The ability to continuously acquire new knowledge and skills is crucial for autonomous agents. Existing methods are typically based on either fixed-size models that struggle to learn a large number of diverse behaviors, or growing-size models that scale poorly with the number of tasks. In this work, we aim to strike a better balance between an agent's *size* and *performance* by designing a method that grows adaptively depending on the task sequence. We introduce Continual Subspace of Policies (CSP), a new approach that incrementally builds a subspace of policies for training a reinforcement learning agent on a sequence of tasks. The subspace's high expressivity allows CSP to perform well for many different tasks while growing sublinearly with the number of tasks. Our method does not suffer from forgetting and displays positive transfer to new tasks. CSP outperforms a number of popular baselines on a wide range of scenarios from two challenging domains, Brax (locomotion) and Continual World (manipulation). Interactive visualizations of the subspace can be found at `csp`. Code is available `here`.

## 1 INTRODUCTION

Developing autonomous agents that can continuously acquire new knowledge and skills is a major challenge in machine learning, with broad application in fields like robotics or dialogue systems. In the past few years, there has been growing interest in the problem of training agents on sequences of tasks, also referred to as continual reinforcement learning (CRL, Khetarpal et al. (2020)). However, current methods either use *fixed-size* models that struggle to learn a large number of diverse behaviors (Hinton et al., 2006; Rusu et al., 2016a; Li & Hoiem, 2018; Bengio & LeCun, 2007; Kaplanis et al., 2019; Traoré et al., 2019; Kirkpatrick et al., 2017; Schwarz et al., 2018; Mallya & Lazebnik, 2018), or *growing-size* models that scale poorly with the number of tasks (**?**Cheung et al., 2019; Wortsman et al., 2020). In this work, we introduce an *adaptive-size* model which strikes a better balance between **performance** and **size**, two crucial properties of continual learning systems (Veniat et al., 2020), thus scaling better to long task sequences.

Taking inspiration from the mode connectivity literature (Garipov et al., 2018; Gaya et al., 2021), we propose **Continual Subspace of Policies** (**CSP**), a new CRL approach that incrementally builds a subspace of policies (see Figure 1 for an illustration of our method). Instead of learning a single policy, CSP maintains an entire subspace of policies defined as a convex hull in parameter space. The vertices of this convex hull are called anchors, with each anchor representing the parameters of a policy. This subspace captures a large number of diverse behaviors, enabling good performance on a wide range of settings. At every stage of the CRL process, the best found policy for a previously seen task is represented as a single point in the current subspace (*i.e.* unique convex combination of the anchors), which facilitates cheap storage and easy retrieval of prior solutions.

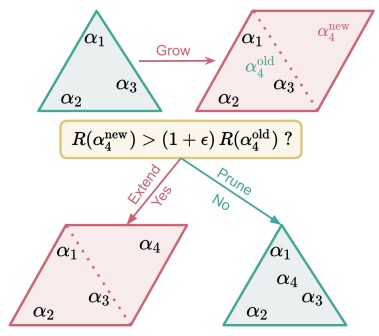
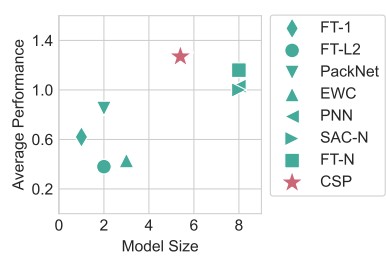

(a) Continual Subspace of Policies (CSP)  (b) Performance-Size Trade-Off

Figure 1: (a) **Continual Subspace of Policies (CSP)** iteratively learns a subspace of policies in the continual RL setting. At every stage during training, the subspace is a simplex defined by a set of anchors (*i.e.* vertices). Any policy (*i.e.* point) in this simplex can be represented as a convex combination $\alpha$ of the anchor parameters. $\alpha_i$ defines the best policy in the subspace for task $i$. When the agent encounters a new task, CSP tentatively *grows* the subspace by adding a new anchor. If the new task $i$ is very different from previously seen ones, a better policy $\alpha_i^{\text{new}}$ can usually be learned in the new subspace. In this case, CSP *extends* the subspace by keeping the new anchor. If the new task bears some similarities to previously seen ones, a good policy $\alpha_i^{\text{old}}$ can typically be found in the old subspace. In this case, CSP *prunes* the subspace by removing the new anchor. The subspace is extended only if it improves performance relative to the old subspace by at least some threshold $\epsilon$. (b) Trade-off between model performance and size for a number of methods, on a sequence of 8 tasks from HalfCheetah (see Table 7 for trade-offs on other scenarios).

If a new task shares some similarities with previously seen ones, a good policy can often be found in the current subspace without increasing the number of parameters. On the other hand, if a new task is very different from previously seen ones, CSP extends the current subspace by adding another anchor, and learns a new policy in the extended subspace. In this case, the pool of candidate solutions in the subspace increases, allowing CSP to deal with more diverse tasks in the future. *The size of the subspace is increased only if this leads to performance gains larger than a given threshold, allowing users to specify the desired trade-off between performance and size* (*i.e.* number of parameters or memory cost).

We evaluate our approach on 18 CRL scenarios from two different domains, locomotion in Brax and robotic manipulation in Continual World, a challenging CRL benchmark (Wołczyk et al., 2021). We also compare CSP with a number of popular CRL baselines, including both fixed-size and growing-size methods. As Figure 1b shows, CSP *is competitive with the strongest existing methods while maintaining a smaller size*. CSP does not incur any forgetting of prior tasks and displays positive transfer to new ones. We demonstrate that by increasing the threshold parameter, the model size can be significantly reduced without substantially hurting performance. In addition, our qualitative analysis shows that the subspace captures diverse behaviors and even combinations of previously learned skills, allowing transfer to many new tasks without requiring additional parameters.

## 2 CONTINUAL REINFORCEMENT LEARNING

A continual RL problem is defined by a sequence of $N$ tasks denoted $t_1, ..., t_N$. Task $i$ is defined by a Markov Decision Process (MDP) $\mathcal{M}_i = \langle \mathcal{S}_i, \mathcal{A}_i, \mathcal{T}_i, r_i, \gamma \rangle$ with a set of states $\mathcal{S}_i$, a set of actions $\mathcal{A}_i$, a transition function $\mathcal{T}_i : \mathcal{S}_i \times \mathcal{A}_i \rightarrow \mathcal{P}(\mathcal{S}_i)$, and a reward function $r_i : \mathcal{S}_i \times \mathcal{A}_i \rightarrow \mathbb{R}$. Here, we consider that all the tasks have the same state and action space. We also define a global policy $\Pi : [1..N] \times \mathcal{Z} \rightarrow (\mathcal{S} \rightarrow \mathcal{P}(\mathcal{A}))$ which takes as input a task id $i$ and a sequence of tasks $\mathcal{Z}$, and outputs the policy which should be used when interacting with task $i$. $\mathcal{P}(\mathcal{A})$ is a probability distribution over the action space. We consider that each task $i$ is associated with a *training budget* of interactions $b_i$. When the system switches to task $t_{i+1}$ it no longer has access to transitions from the $\mathcal{M}_i$. We do not assume access to a replay buffer with transitions from prior tasks since this would lead to a significant increase in memory cost; instead, at each stage, we maintain a buffer with transitions from the current task, just like SAC (Haarnoja et al., 2018b).

## 3 CONTINUAL SUBSPACE OF POLICIES (CSP)

### 3.1 SUBSPACE OF POLICIES

Our work builds on Gaya et al. (2021) by leveraging a subspace of policies. However, instead of using the subspace to train on a single environment and adapt to new ones at test time, we use it to efficiently learn tasks sequentially in the continual RL setting. This requires designing a new approach for learning the subspace, as detailed below.

Let $\theta \in \mathbb{R}^d$ represent the parameters of a policy, which we denote as $\pi(a|s, \theta)$. A subspace of policies is a simplex in $\mathbb{R}^d$ defined by a set of anchors $\Theta = \{\theta_1, ..., \theta_k\}$, where any policy in this simplex can be represented as a convex combination of the anchor parameters. Let the weight vector $\alpha \in \mathbb{R}^k_+$ with $\|\alpha\|_1 = 1$ denote the coefficients of the convex combination. Therefore, each value of $\alpha$ uniquely defines a policy with $\pi(a|s, [\alpha, \Theta]) = \pi(a|s, \sum \alpha_i \theta_i)$. Throughout the paper, we sometimes refer to the weight vector $\alpha$ and its corresponding policy $\pi(a|s, [\alpha, \Theta])$ interchangeably.

### 3.2 LEARNING ALGORITHM

We propose to adaptively construct a subspace of policies for the continual RL setting in which an agent learns tasks sequentially. We call our method **Continual Subspace of Policies** (**CSP**). A pseudo-code is available in Appendix C.1.

Our model builds a sequence of subspaces $\Theta_1, ..., \Theta_N$, one new subspace after each training task, with each subspace extending the previous one. Note that each subspace $\Theta_j$ is a collection of at most $j$ anchors (or neural networks) *i.e.* $|\Theta_j| \leq j$, $\forall \; j \in 1...N$ since the subspace grows sublinearly with the number of tasks (as explained below). Hence, the number of anchors $m$ of a subspace $\Theta_j$ is not the same as the number of tasks $j$ used to create $\Theta_j$. The learned policies are represented as single points in these subspaces. At each stage, CSP maintains both a set of anchors defining the current subspace, as well as the weights $\alpha$ corresponding to the best policies found for all prior tasks. The best found policy for task $i$ after training on the first $j$ tasks, $\forall \; i \leq j$, is denoted as $\pi_i^j(a|s)$ and can be represented as a point in the subspace $\Theta_j$ with a weight vector denoted $\alpha_i^j$ such that $\pi_i^j(a|s) = \pi(a|s, [\alpha_i^j, \Theta_j])$, where $|\alpha_i^j| = |\Theta_j| \leq j$.

Given a set of anchors $\Theta_j$ and a set of previously learned policies $\{\alpha_1^j, ..., \alpha_j^j\}$, updating our model on the new task $t_{j+1}$ produces a new subspace $\Theta_{j+1}$ and a new set of weights $\{\alpha_1^{j+1}, ..., \alpha_{j+1}^{j+1}\}$. There are two possible cases. One possibility is that the current subspace already contains a good policy for $t_{j+1}$, so we just need to find the weight vector corresponding to a policy which performs well on $t_{j+1}$. The other possibility is that the current subspace does not contain a good policy for $t_{j+1}$. In this case, the algorithm produces a new subspace by adding one anchor to the previous one (see next section), and converts the previous policies to be compatible with this new subspace.

To achieve this, CSP operates in two phases:

1. *Grow* the current subspace by adding a new anchor and learning the best possible policy for task $j + 1$ in this subspace (where the previous $j$ anchors are frozen).

2. Compare the quality of this policy with the best possible policy expressed in the previous subspace. Based on this comparison, decide whether to *extend* the subspace to match the new one or *prune* it back to the previous one.

We now describe in detail the two phases of the learning process, namely how we grow the subspace and how we decide whether to extend or prune it (see Algorithm 1 for a pseudo-code).

### 3.3 GROW THE SUBSPACE

Given the current subspace $\Theta_j$ composed of $m \leq j$ anchors (with $j$ being the number of tasks seen so far), a new subspace $\tilde{\Theta}_{j+1}$ is built as follows. First a new anchor denoted $\theta_{j+1}$ is added to the set of anchors such that $\tilde{\Theta}_{j+1} = \Theta_j \bigcup \{\theta_{j+1}\}$. With all previous anchors frozen, we train the new anchor by sampling $\alpha$ values from a Dirichlet distribution parameterized by a vector with size $j + 1$, $Dir(\mathcal{U}(j + 1))$. The new anchor $\theta_{j+1}$ is updated by maximizing the expected return obtained by interacting with the new task $j + 1$ using policies defined by the sampled $\alpha$'s:

$$\theta_{j+1} = \arg\max_{\theta} \mathbb{E}_{\alpha \sim Dir, \tau \sim \pi(a|s, [\alpha, \Theta_j \bigcup \{\theta\}])} [R_{j+1}(\tau)] \tag{1}$$

where $R_{j+1}(\tau)$ is the return obtained on task $j + 1$ throughout trajectory $\tau$ which was generated using policy $\pi(a|s, [\alpha, \tilde{\Theta}_{j+1}])$. Note that the anchor is trained such that not only one but all possible values of $\alpha$ tend to produce a good policy. To do so, we sample different $\alpha$ per episode. The resulting subspace thus aims at containing as many good policies as possible for task $j + 1$.

### 3.4 EXTEND OR PRUNE THE SUBSPACE

To decide if the new anchor is kept, we propose to simply compare the best possible policy for task $j + 1$ in the new subspace with the best possible policy for the same task in the previous subspace (*i.e.* without using the new anchor). Each policy could be evaluated via Monte-Carlo (MC) estimates by doing additional rollouts in the environment and recording the average performance. However, this typically requires a large number (*e.g.* millions) of interactions which may be impractical with a limited budget. Thus, we propose an alternative procedure to make this evaluation sample efficient.

For each task $j$ and corresponding subspace $\Theta_j$, our algorithm also learns a $Q$-function $Q(s, a, \alpha)$ which is trained to predict the expected return on task $j$ for all possible states, actions, and all possible $\alpha$'s in the corresponding subspace. This $Q$-function is slightly different from the classical ones in RL since it takes as an additional input the vector $\alpha$. This $Q$-function is reset for every new task. Our algorithm is based on SAC (Haarnoja et al., 2018b), so for each task, we collect a replay buffer of interactions $\mathcal{B}$ which contains all states and actions seen by the agent while training on that task. Thus, the $Q$-function $Q(s, a, \alpha)$ can help us directly estimate the quality $W(\alpha)$ of the policy represented by the weight vector $\alpha$ in the new subspace which can be computed as the average over all states and actions in the replay buffer:

$$W(\alpha) = \mathbb{E}_{s,a \sim \mathcal{B}} Q(s, a, \alpha). \tag{2}$$

It is thus possible to compute the value of $\alpha$ corresponding to the best policy in the extended subspace (denoted $\alpha^{\text{new}} \in \mathbb{R}_+^{m+1}$, $\|\alpha^{\text{new}}\|_1 = 1$):

$$\alpha^{\text{new}} = \underset{\alpha \in \mathbb{R}_+^{m+1}, \|\alpha\|_1 = 1}{\arg\max} W(\alpha), \tag{3}$$

as well as the value of $\alpha$ corresponding to the best policy in the previous subspace (denoted $\alpha^{\text{old}} \in \mathbb{R}_+^m$, $\|\alpha^{\text{old}}\|_1 = 1$):

$$\alpha^{\text{old}} = \underset{(\alpha, 0) \text{ with } \alpha \in \mathbb{R}_+^m, \|\alpha\|_1 = 1}{\arg\max} W(\alpha). \tag{4}$$

In practice, $\alpha^{\text{new}}$ and $\alpha^{\text{old}}$ are estimated by uniformly sampling a number of $\alpha$'s in the corresponding subspace as well as a number of states and actions from the buffer.

The quality of the new subspace and the previous one can thus be evaluated by comparing $W(\alpha^{\text{new}})$ and $W(\alpha^{\text{old}})$. If $W(\alpha^{\text{new}}) > (1 + \epsilon) \cdot W(\alpha^{\text{old}})$, the subspace is extended to the new subspace (*i.e.* the one after the grow phase): $\Theta_{j+1} = \tilde{\Theta}_{j+1}$. Otherwise, the subspace is pruned back to the old subspace (*i.e.* the one before the growth phase): $\Theta_{j+1} = \Theta_j$. Note that, if the subspace is extended, the previously learned policies have to be mapped in the new subspace such that $\alpha_i^{j+1} \in \mathbb{R}_+^{j+1}$ *i.e.* $\forall i \leq j$, $\alpha_i^{j+1} := (\alpha_i^j, 0)$ and $\alpha_{j+1}^{j+1} := \alpha^{\text{new}}$. If the subspace is not extended, then old values can be kept *i.e.* $\forall i \leq j$, $\alpha_i^{j+1} := \alpha_i^j$ and $\alpha_{j+1}^{j+1} := \alpha^{\text{old}}$.

After finding the best $\alpha$ for the current task, the replay buffer and Q-function are reinitialized. Hence, the memory cost for training the Q-function is constant and there is no need to store data from previous tasks unlike other CRL approaches (Rolnick et al., 2019). The policy $\pi_{\theta_{j+1}}$ and value function $W_\phi$ are updated using SAC (Haarnoja et al., 2018b). See Appendix C for more details.

### 3.5 SCALABILITY OF CSP

By having access to an large number of policies, the subspace is highly expressive so it can capture a wide range of diverse behaviors. This enables positive transfer to many new tasks without the need for additional parameters. As a consequence, *the number of parameters increases sublinearly with the the number of tasks*. The final size of the subspace depends on the sequence of tasks, with longer and more diverse sequences requiring more anchors. The speed of growth is controlled by the threshold $\epsilon$ which trades-off performance gains for memory efficiency (the higher the $\epsilon$ the more performance losses are tolerated to reduce memory costs). See Figure 2a and Appendix C.2.

## 4 EXPERIMENTS

### 4.1 ENVIRONMENTS

We evaluate CSP on 18 CRL scenarios containing 35 different RL tasks, from two continuous control domains, *locomotion* in **Brax** (Freeman et al., 2021) and *robotic manipulation* in **Continual World** (CW, Wołczyk et al. (2021)), a challenging CRL benchmark. For CW, we run experiments on both the proposed sequence of 10 tasks (CW10), and on all 8 triplet sequences (CW3). Each task in these sequences has a *different reward* function. The goal of these experiments is to compare our approach with popular CRL methods on a well-established benchmark.

For Brax, we create new CRL scenarios based on 3 subdomains: **HalfCheetah**, **Ant**, and **Humanoid**. Each scenario has 8 tasks and each task has *specific dynamics*. We use a budget of 1M interactions for each task. The goal of these experiments is to perform an in-depth study to separately evaluate capabilities specific to CRL agents such as *forgetting, transfer, robustness, and compositionality* (see Appendix B.2 and Tables 5a, 5b for more information). For each of the 4 capabilities, we create 2 CRL scenarios, one based on HalfCheetah and one based on Ant. To further probe the effectiveness of our approach, we also create one CRL scenario with 4 tasks on the challenging Humanoid domain. Here we use a budget of 2M interactions for each task.

The CRL scenarios are created following the protocol introduced in Wołczyk et al. (2021) which proposes a systematic way of generating task sequences that test CRL agents along particular axes. While Wołczyk et al. (2021) focus on transfer and forgetting (see Appendix B.3), we also probe robustness (*e.g.* adapting to environment perturbations such as action reversal), and compositionality (*e.g.* combining two previously learned skills to solve a new task). Each task in a sequence has different dynamics which are grounded in quasi-realistic situations such as increased or decreased gravity, friction, or limb lengths. See Appendix B.1 for details on how these tasks were designed.

### 4.2 BASELINES

We compare CSP with a number of popular CRL baselines such as PNN (Rusu et al., 2016b), EWC (Kirkpatrick et al., 2017), PACKNET (Mallya & Lazebnik, 2018), FT-1 which finetunes a single model on the entire sequence of tasks, and FT-L2 which is like FT-1 with an additional $L_2$ regularization applied during finetuning. We also compare with SAC-N which trains one model for each task from scratch. While SAC-N avoids forgetting, it cannot transfer knowledge across tasks. Finally, we compare with a method called FT-N which combines the best of both SAC-N and FT-1. Just like SAC-N, it stores one model per task after training on it and just like FT-1, it finetunes the previous model to promote transfer. However, FT-N and SAC-N scale poorly (*i.e.* linearly) with the number of tasks in terms of both memory and compute, which makes them unfeasible for real-world applications. Note that our method is not directly comparable with CLEAR (Rolnick et al., 2019) since we assume no access to data from prior tasks. Storing data from all prior tasks (as CLEAR does) is unfeasible for long task sequences due to prohibitive memory costs. All methods use SAC (Haarnoja et al., 2018b) as a base algorithm. The means and standard deviations are computed over 10 seeds unless otherwise noted. See Appendices A.1, A.2, and A.4 for more details about our protocol, baselines, and hyperparameters, respectively.

### 4.3 ABLATIONS

We also perform a number of ablations to understand how the different components of CSP influence performance. We first compare CSP with CSP-ORACLE which selects the best policy by sampling a large number of policies in the subspace and computing Monte-Carlo estimates of their returns. These estimates are expected to be more accurate than the critic's, so CSP-ORACLE can be considered an upper bound to CSP. However, CSP-ORACLE is less efficient than CSP since it requires singificantly more interactions with the environment to find a policy for each task (*i.e.* millions).

We also want to understand how much performance we lose, if any, by not adding one anchor per task. To do this, we run an ablation with no threshold which always extends the subspace by adding one anchor for each new task. In addition, we vary the threshold $\epsilon$ used to decide whether to extend the subspace based on how much performance is gained by doing so. This analysis can shed more light on the trade-off between performance gain and memory cost as the threshold varies.

Table 1: Aggregated results across all Brax scenarios from HalfCheetah, Ant, and Humanoid. These scenarios were designed to test forgetting, transfer, compositionality, and robustness. CSP performs as well as or better than the strongest baselines, while having a much lower model size and thus memory cost. CSP's performance is also not too far from that of CSP-ORACLE indicating that it can use the critic to find good policies in the subspace without requiring millions of interactions.

| Method | HalfCheetah (4 scenarios) | | Ant (4 scenarios) | | Humanoid (1 scenario) | |
|---|---|---|---|---|---|---|
| | Performance | Model Size | Performance | Model Size | Performance | Model Size |
| FT-1 | $0.62 \pm 0.29$ | 1.0 | $0.52 \pm 0.26$ | 1.0 | $0.71 \pm 0.07$ | 1.0 |
| FT-L2 | $0.38 \pm 0.15$ | 2.0 | $0.78 \pm 0.20$ | 2.0 | $0.68 \pm 0.28$ | 2.0 |
| PACKNET | $0.85 \pm 0.14$ | 2.0 | $1.08 \pm 0.21$ | 2.0 | $0.96 \pm 0.21$ | 2.0 |
| EWC | $0.43 \pm 0.24$ | 3.0 | $0.55 \pm 0.24$ | 3.0 | $0.94 \pm 0.01$ | 3.0 |
| PNN | $1.03 \pm 0.14$ | 8.0 | $0.98 \pm 0.31$ | 8.0 | $0.98 \pm 0.26$ | 4.0 |
| SAC-N | $1.00 \pm 0.15$ | 8.0 | $1.00 \pm 0.38$ | 8.0 | $1.00 \pm 0.29$ | 4.0 |
| FT-N | $1.16 \pm 0.20$ | 8.0 | $0.97 \pm 0.20$ | 8.0 | $0.65 \pm 0.46$ | 4.0 |
| CSP (ours) | $\mathbf{1.27 \pm 0.27}$ | $5.4 \pm 1.3$ | $\mathbf{1.11 \pm 0.17}$ | $3.9 \pm 0.8$ | $\mathbf{1.76 \pm 0.19}$ | $3.4 \pm 0.3$ |
| CSP-ORACLE | $1.88 \pm 0.19$ | | $1.24 \pm 0.07$ | | $1.98 \pm 0.22$ | |

## 4.4 METRICS

Agents are evaluated across a range of commonly used CRL metrics (Wołczyk et al., 2021; Powers et al., 2021). For brevity, following the notation in Section 2, we will use $\pi_i^j \coloneqq \Pi(i, [t_1, \ldots, t_j])$ to denote the policy selected by $\Pi$ for task $i$ obtained after training on the sequence of tasks $t_1, \ldots, t_j$ (in this order with $j$ included). Note that $\Pi$ can also be defined for a single task *i.e.* $\pi_i^{t_i} \coloneqq \Pi(i, [t_i])$ is the policy for task $i$ after training the system only on task $t_i$ (for the corresponding budget $b_i$). We also report forgetting and forward transfer (see Appendix A.3 for their formal definitions).

**Average Performance** is the average performance of the method across all tasks, after training on the entire sequence of tasks. Let $P_i(\pi)$ be the performance of the system on task $i$ using policy $\pi$ defined as $P_i(\pi) \coloneqq \mathbb{E}_{\pi, \mathcal{T}_i} \left[ \sum r_i(s, a) \right]$, where $r_i(s, a)$ is the reward on task $i$ when taking action $a$ in state $s$. Then, the final average performance across all tasks can be computed as $P \coloneqq \frac{1}{N} \sum_{i=1}^N P_i(\pi_i^N)$.

**Model Size** is the final number of parameters of a method after training on all tasks, divided by the number of parameters of FT-1 (which is a single policy trained on all tasks, with size equal to that of an anchor). For example, in the case of CSP, the model size is equivalent with the number of anchors defining the final subspace $|\Theta_N|$. Similarly, in the case of FT-N and SAC-N, the model size is always equal to the number of tasks N. Following Rusu et al. (2016b), we cap PNN's model size to N since otherwise it grows quadratically with the number of tasks.

## 5 RESULTS

### 5.1 PERFORMANCE ON BRAX

Table 1 shows the aggregated results across all scenarios from HalfCheetah, Ant, and Humanoid. The results are normalized using SAC-N's performance. See Appendix D.1 for individual results.

CSP performs as well as or better than the strongest baselines which grow linearly with the number of tasks (*i.e.* PNN, SAC-N, FT-N), while maintaining a much smaller size and thus memory cost. In fact, CSP's size grows sublinearly with the number of tasks. At the same time, CSP is significantly better than the most memory-efficient baselines that have a relatively small size (*i.e.* FT-1, FT-L2, EWC, PACKNET). Naive methods like FT-1 have low memory costs and good transfer, but suffer from catastrophic forgetting. In contrast, methods that aim to reduce forgetting such as FT-L2 or EWC do so at the expense of transfer. The only competitive methods in terms of performance are the ones where the number of parameters increases at least linearly with the number of tasks, such as PNN, SAC-N, and FT-N. These methods have no forgetting because they store the models trained on each task. SAC-N has no transfer since it trains each model from scratch, while FT-N promotes transfer as it finetunes the previous model. However, due to their poor scalability, these methods are unfeasible for more challenging CRL scenarios with long and varied task sequences.

Table 2: Results on the CW10 benchmark for CSP and other popular baselines including the state-of-the-art PACKNET (results taken from Wołczyk et al. (2021)). CSP performs almost as well as PACKNET while using about half the number of heads, and thus having a much lower memory cost.

| Method | CSP (ours) | PACKNET | EWC | FT-L2 | FT-1 |
|---|---|---|---|---|---|
| **Performance** | $0.81 \pm 0.06$ | **$0.83 \pm 0.02$** | $0.66 \pm 0.03$ | $0.48 \pm 0.05$ | $0.10 \pm 0.01$ |
| **# Heads** | **$5.3 \pm 1.6$** | 10 | 10 | 10 | 10 |

In contrast, CSP doesn't suffer from forgetting since the best found policies for all prior tasks can be cheaply stored in the form of convex combinations of the subspace's anchors (*i.e.* vectors rather than model weights). In addition, due to having access to a large number of policies (*i.e.* all convex combinations of the anchors), CSP has good transfer to new tasks (that share some similarities with prior ones). This allows CSP to achieve strong performance while its memory cost grows sublinearly with the number of tasks. Nevertheless, CSP's performance is not too far from that of CSP-ORACLE, indicating that the critic can be used to find good policies in the subspace without requiring millions of interactions. See Appendix C.3 for more details about the use of the critic.

## 5.2 PERFORMANCE ON CONTINUAL WORLD

Table 2 shows results on CW10 (Wołczyk et al., 2021), a popular CRL benchmark. The baselines used in Wołczyk et al. (2021) use a separate linear layer for each task on top of a common network. The authors recognize this as a limitation since it scales poorly with the number of tasks. For a fair comparison, we implement CSP using a similar protocol where the anchors are represented by linear heads, so the number of heads grows adaptively depending on the performance threshold $\epsilon$. Instead of model size, we compare the final number of heads, since this now determines a method's scalability with the number of tasks. See Appendix A for more details about the experimental setup.

CSP performs almost as well as PACKNET which is the current state-of-the-art on CW10 (Wołczyk et al., 2021), and is significantly better than all other baselines. At the same time, CSP uses about half the number of heads, thus being more memory efficient especially as the number of tasks increases. Note that PACKNET suffers from a major limitation, namely that it requires prior knowledge of the total number of tasks in order to allocate resources. If this information is not correctly specified, PACKNET is likely to fail due to either not being expressive enough to handle many tasks or being too inefficient while learning only a few tasks (Wołczyk et al., 2021). This makes PACKNET unfeasible for real-world applications where agents can face task sequences of varying lengths (including effectively infinite ones). In contrast, CSP grows adaptively depending on the sequence of tasks, so it can handle both short and long sequences without any modification to the algorithm. See Appendix D.2 for additional results on Continual World, including CW3.

To summarize, CSP *is competitive with the strongest CRL baselines, while having a lower memory cost since its size grows sublinearly with the number of tasks. Thus,* CSP *maintains a good balance between model size and performance, which allows it to scale well to long task sequences.*

## 5.3 ANALYSIS OF CSP

**Varying the Threshold.** Figure 2a shows how performance and size vary with the threshold $\epsilon$ used to decide whether to extend the subspace or not. Both the performance and the size are normalized with respect to CSP-LINEAR which always extends the subspace. As expected, as $\epsilon$ increases, performance decreases, but so does the size. Note that *performance is still above* 70% *even as the size is cut by more than* 70%, relative to CSP-LINEAR which trains a new set of parameters (*i.e.* anchor) for each task. Thus, CSP *can drastically reduce memory costs without significantly hurting performance*. Practitioners can set the threshold to trade-off performance gains for memory efficiency *i.e.* the higher the $\epsilon$ the more performance losses are tolerated in order to reduce memory costs. In practice, we found $\epsilon = 0.1$ to offer a good trade-off between performance and model size.

**Scalability.** Figure 2b shows how performance and size scale with the number of tasks in the sequence (on HalfCheetah's compositionality scenario) for both CSP and FT-N which is our strongest baseline on this scenario. CSP maintains both strong performance and small size (*i.e.* low memory cost) even as the number of tasks increases. In contrast, even if FT-N performs well on all these scenarios, its size grows linearly with the number of tasks, rendering it impractical for long task sequences.

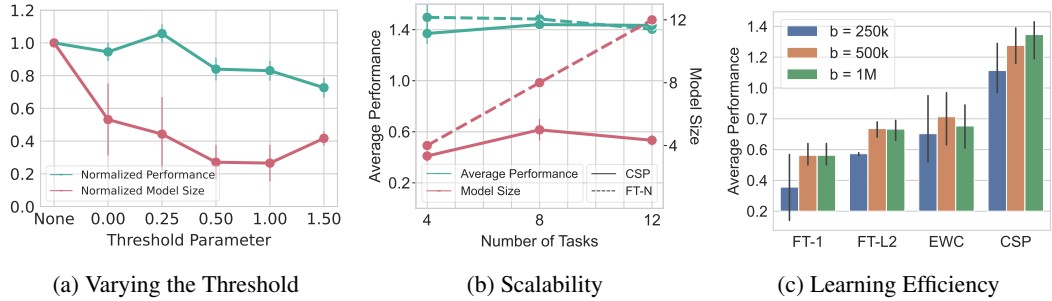

|  |  |  |
|---|---|---|
| (a) Varying the Threshold | (b) Scalability | (c) Learning Efficiency |

Figure 2: (a) shows how CSP's performance and size vary with the threshold, (b) shows how performance and size scale with the number of tasks for CSP and FT-N, and (c) compares the performance of multiple methods using different budgets. See Appendix D.3 for more details.

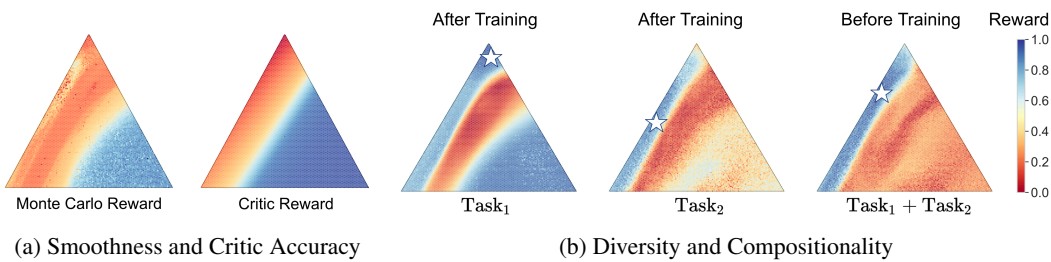

|  |  |
|---|---|
| (a) Smoothness and Critic Accuracy | (b) Diversity and Compositionality |

Figure 3: (a) shows the value of each policy in the subspace estimated using both Monte-Carlo simulations (left) and our critic's predictions (right), demonstrating both that the subspace is smooth and that the critic learns accurate estimates of the reward, which allows CSP to find good policies in the subspace; (b) shows the subspace for three different tasks, the third being a combination of the first two (*e.g.* walk on the moon (task 1) with tiny feet (task 2)), demonstrating that the subspace already contains a policy with high reward on the compositional task before being trained on it. The star represents the best policy in the subspace for the corresponding task. The subspace contains policies covering the whole spectrum of rewards, suggesting that it captures diverse behaviors. Click here to see interactive visualizations of the subspace.

**Learning Efficiency.** Figure 2c shows the average performance for three different budgets (*i.e.* number of interactions allowed for each task) on HalfCheetah's robustness scenario, comparing CSP with FT-1, FT-L2, and EWC. These results demonstrate that CSP can learn efficiently even with a reduced budget, while still outperforming these baselines. By keeping track of all convex combinations of previously learned behaviors, CSP enables good transfer to new tasks which in turn leads to efficient training on task sequences.

## 5.4 ANALYSIS OF THE SUBSPACE

**Smoothness and Critic Accuracy.** Figure 3a shows a snapshot of a trained subspace, along with the expected reward of all policies in the subspace, for a given task. The expected reward is computed using both Monte-Carlo (MC) rollouts, as well as our critic's predictions using Equation 2. As illustrated, the learned Q-function has a similar landscape with the MC reward *i.e.* the subspace is smooth and the critic's estimates are accurate.

**Diversity and Compositionality.** Figure 3b illustrates that the subspace contains behaviors composed of previously learned skills (*e.g.* walk on the moon, walk with tiny feet, walk on the moon with tiny feet). This allows CSP to reuse previously learned skills to find good policies for new tasks without the need for additional parameters. The figure also shows that for a given task, the policies in the subspace cover the whole spectrum of rewards, thus emphasizing the diversity of the behaviors expressed by the subspace. See Appendix C.6 for more analysis of the subspace.

## 6 RELATED WORK

**Continual Reinforcement Learning.** CRL methods aim to avoid catastrophic forgetting, as well as enable transfer to new tasks, while remaining scalable to a large number of tasks. In the past few years, multiple approaches have been proposed to address one or more of these challenges (Tessler et al., 2017; Parisi et al., 2019; Khetarpal et al., 2020; Powers et al., 2021; Wołczyk et al., 2021; Tessler et al., 2017; Berseth et al., 2022; Nagabandi et al., 2018; Xie et al., 2020; Xu et al., 2020; Berseth et al., 2021; Ren et al., 2022; Kessler et al., 2022).

One class of methods focuses on preventing catastrophic *forgetting*. Some algorithms achieve this by storing the parameters of models trained on prior tasks (Rusu et al., 2016b; Cheung et al., 2019; Wortsman et al., 2020). However, these methods scale poorly with the number of tasks (*i.e.* at least linearly) in both compute and memory, which makes them unfeasible for more realistic scenarios. Among them, PACKNET (Mallya & Lazebnik, 2018) performs well on several benchmarks (CW10, CW20) but requires the final number of tasks as input. An extension of this method called EfficientPackNet is proposed by Schwarz et al. (2021) to overcome this issue, but it is only evaluated on a subset of CW. Other methods maintain a buffer of experience from prior tasks to alleviate forgetting (Lopez-Paz & Ranzato, 2017; Isele & Cosgun, 2018; Riemer et al., 2019b; Rolnick et al., 2019; Caccia et al., 2020a). However, this is also not scalable as the memory cost increases significantly with the number and complexity of the tasks. In addition, many real-world applications in domains like healthcare or insurance prevent data storage due to privacy or ethical concerns. Another class of methods focuses on improving *transfer* to new tasks. Naive approaches like finetuning that train a single model on each new task provide good scalability and plasticity, but suffer from catastrophic forgetting. To overcome this effect, methods like elastic weight consolidation (EWC ) (Kirkpatrick et al., 2017) alleviate catastrophic forgetting by constraining how much the network's weights change, but this can in turn reduce plasticity. Another class of methods employs knowledge distillation to improve transfer in CRL (Hinton et al., 2006; Rusu et al., 2016a; Li & Hoiem, 2018; Schwarz et al., 2018; Bengio & LeCun, 2007; Kaplanis et al., 2019; Traoré et al., 2019). However, since these methods train a single network, they struggle to capture a large number of diverse behaviors.

There is also a large body of work which leverages the shared structure of the tasks (Xu & Zhu, 2018; Pasunuru & Bansal, 2019; Lu et al., 2021; Mankowitz et al., 2018; Abel et al., 2017; Sodhani et al., 2021), meta-learning (Javed & White, 2019; Spigler, 2019; Beaulieu et al., 2020; Caccia et al., 2020b; Riemer et al., 2019a; Zhou et al., 2020; Schmidhuber, 2013), or generative models (Robins, 1995; Silver et al., 2013; Shin et al., 2017; Atkinson et al., 2021) to improve CRL agents, but these methods don't work very well in practice. Lee et al. (2020) use an iteratively expanding network but they assume no task boundaries and only consider the supervised learning setting.

**Mode Connectivity and Neural Network Subspaces.** Based on prior studies of mode connectivity (Garipov et al., 2018; Draxler et al., 2018; Mirzadeh et al., 2021), Wortsman et al. (2021); Benton et al. (2021) proposed the neural network subspaces to connect the solutions of different models. More similar to our work, Doan et al. (2022) leverages subspace properties to mitigate forgetting on a sequence of supervised learning tasks, but doesn't focus on other aspects of the continual learning problem. Our work was also inspired by Gaya et al. (2021) which learns a subspace of policies for a single task for fast adaptation to new environment. In contrast to Gaya et al. (2021), we consider the continual RL setting and adaptively grow the subspace as the agent encounters new tasks. Our work is first to demonstrate the effectiveness of mode connectivity for the continual RL setting.

## 7 DISCUSSION

In this paper, we propose CSP, a new continual RL method which adaptively builds a subspace of policies to learn a sequence of tasks. CSP is competitive with the best existing methods while using significantly fewer parameters. Thus, it strikes a good balance between performance and memory cost which allows it to scale well to long task sequences. Our paper is first to use a subspace of policies for continual RL, and thus opens up many interesting directions for future work. For example, one can assume a fixed size for the subspace and update all the anchors whenever the agent encounters a new task. Another promising direction is to leverage the structure of the subspace to meta-learn or search for good policies on a given task. Finally, active learning techniques could be employed to more efficiently evaluate policies contained in the subspace.

## 8 REPRODUCIBILITY STATEMENT

This work builds on the open-source RL library SaLinA (Denoyer et al., 2021) and we have provided a detailed description of the experiments, environments, and our method in Appendix A, B, and C respectively. Code is available here.

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

# A  EXPERIMENTAL DETAILS

All the experiments were implemented with SaLinA (**?**), a flexible and simple python library for learning sequential agents. We used Soft Actor Critic (Haarnoja et al., 2018b) as the routine algorithm for each method. It is a popular and efficient off-policy algorithm for continuous domains. We use a version where the entropy parameter is learned, as proposed in Haarnoja et al. (2018a). For each task and each methods, the twin critic networks are reset as well as the replay buffer (which has a maximum size of 1M interactions). This is a common setting, used by several CRL benchmarks (Wołczyk et al., 2021; Powers et al., 2021)

## A.1  EXPERIMENTAL PROTOCOL

**Brax scenarios** Considering FT-N as the upper bound with which we want to compare, we decided to apply the following protocol for each short scenario. First, we run a gridsearch on SAC hyper-parameters (see Table 3) on FT-N and select the best set in terms of final average performance (see Section 4 for the details about this metric). Then, we freeze these hyper-parameters and performed a specific gridsearch for CSP and each baseline (see Table 4). Each hyper-parameter set is evaluated over 10 seeds. We believe this ensures fair comparisons, and even gives a slight advantage to FT-N compared to our method. Note that we set the number of parallel environments to 128 and the number of update per step to 0.5. They can be seen as tasks constraints. These values are reasonable, and FineTuning them would have increased the number of experiments by a lot. Evaluations are done on 512 environments per experiment, each with different seeds, with no noise on the policy output.

**Continual World** We bypassed the preliminary SAC gridsearch and use the set of hyperparameters proposed in Wołczyk et al. (2021).For the triplet scenarios, we performed a specific gridsearch for CSP and each baseline (see Table 4). For CW10, we added CSP framework only on the top layer (head), the backbone being handled by PACKNET. In Wołczyk et al. (2021), PACKNET uses one head per task, just like our baseline FT-N does. In other terms, one can see it as a benchmark between FT-N and CSP on top of PACKNET. Note that we did not performed a gridsearch on any hyperparameter in this case.

## A.2  ARCHITECTURE AND BASELINES

**Architecture for Brax** For the twin critics and policy, We use a network with 4 linear layers, each consisting of 256 neurons. We use leaky ReLU activations (with $\alpha = 0.2$) after every layer.

**Architecture for Continual World** We use the same architecture as Wołczyk et al. (2021) for each method: we add Layer Normalization Ba et al. (2016) after the first layer, followed by a tanh activation.

**EWC** (Kirkpatrick et al., 2017) This regularization based method aims to protect parameters that are important for the previous tasks. After each task, it uses the Fisher information matrix to approximate the importance of each weight. We followed Wołczyk et al. (2021) to compute the Fisher matrix and use an analytical derivation of it.

**FT-L2** This baseline is proposed by Kirkpatrick et al. (2017) and re-used by Wołczyk et al. (2021). It can be seen as a simplified version of EWC where the regularization coefficients for each parameters are equal to 1.

**PNN** (Rusu et al., 2016b) This method creates a new network at the beginning of each task, as well as lateral networks that will take as inputs - for each hidden layer - the output of the networks trained on former tasks. In this method, the number of parameters, training and inference times are growing quadratically with respect to the number of tasks, such that for our scenarios, the vanilla model size would have been around 40. To overcome that, we followed author's advice and reduced the number of hidden layers with respect to the final number of tasks such that it is no bigger than FT-N. Note that, as for PACKNET, this also requires to know this number in advance.

PACKNET (Mallya & Lazebnik, 2018) the leading method in Continual World benchmark aims to learn on a new task and prune the networks in the end, and retrain the weights that have the highest amplitude. This has two drawbacks : one has to allocate in advance a certain number of weight per task. Without additional knowledge on the sequence, authors recommend to evenly split the number of weights per task, meaning that one has to know the final number of tasks. In addition, the training procedure after having pruned the network does not require any interaction with the environment but is computationally expensive. For completeness, we also report the results for the same model with hidden size doubled in Halfcheetah scenarios (see Table 8). We named it PACKNETX2 . The slight increase of performance (7%) is not sufficient to outperform CSP on these scenarios.

## A.3    ADDITIONAL METRICS

Note that for Brax, performance is always normalized using a reference cumulative reward (see Table 11). In Tables 8, 9, 10, 12 we also report two additional classical metrics from continual learning:

**Forward Transfer** measures how much a CRL system is able to transfer knowledge from task to task. At task $i$, it compares the performance of the of the system trained on all previous tasks $t_1, ..., t_i$ to the same model trained solely on task $t_i$. This measure is defined as $T_i := P_i(\pi_i^i) - P_i(\pi_i^{t_i})$, and thus the forward transfer for all tasks is $T := \frac{1}{N} \sum_{i=1}^{N} T_i$.

**Forgetting** evaluates how much a system has forgotten about task $i$ after training on the full sequence of tasks. It thus compares the performance of policy $\pi_i^i$ with the performance of policy $\pi_i^N$ and is defined as $F_i := P_i(\pi_i^i) - P_i(\pi_i^N)$. Similarly to the average transfer, we report the average forgetting across all tasks $F := \frac{1}{N} \sum_{i=1}^{N} F_i$.

## A.4    HYPERPARAMETER SELECTION

Table 3: Hyper-parameters search for SAC over Brax scenarios. The asterix indicates that the hyper-parameter is seen as a constraint of the environment, as explained in A.1.

| Hyper-parameter | Values tested |
|---|---|
| lr policy | $\{0.0003, 0.001\}$ |
| lr critic | $\{0.0003, 0.001\}$ |
| reward scaling | $\{1., 10.\}$ |
| target output std | $\{0.05, 0.1\}$ |
| policy update delay | $\{2, 4\}$ |
| target update delay | $\{2, 4\}$ |
| lr entropy | $0.0003$ |
| update target network coeff | $0.005$ |
| batch size | $256$ |
| n parallel environments | $128^*$ |
| gradient update per step | $0.5^*$ |
| discount factor | $0.99$ |
| replay buffer size | $1M$ |
| warming steps (random uniform policy) | $12,800$ |

Table 4: Specific hyper-parameter search for regularization based baselines and our model.

| Hyper-parameter | Value |
|---|---|
| FT-L2, EWC | |
| regularization coefficient | $\{10^{-2}, 10^0, 10^2, 10^4, 10^6\}$ |
| CSP | |
| threshold | $\{0.1, 0.25\}$ |
| combination rollout length | $\{20, 100\}$ |

## A.5    COMPUTE RESOURCES

Each algorithm was trained using one Intel(R) Xeon(R) CPU cores (E5-2698 v4 @ 2.20GHz) and one NVIDIA V100 GPU. Each run took between approximately 10 and 30 hours to complete for Brax and 50 to 70 hours for CW10. The total runtime depended on three factors: the domain, the computation time of the algorithm and the behavior of the policy.

Table 5: Hyper-parameters selected for each Brax scenario based on FT-N performance as explained in A.1

| Hyper-parameter | Halfcheetah | | | |
|---|---|---|---|---|
| | Forgetting | Transfer | Distraction | Compositional |
| lr policy | 0.001 | 0.0003 | 0.001 | 0.0003 |
| lr critic | 0.0003 | 0.0003 | 0.001 | 0.0003 |
| reward scaling | 1. | 1. | 1. | 10. |
| target output std | 0.1 | 0.05 | 0.1 | 0.1 |
| policy update delay | 2 | 2 | 4 | 4 |
| target update delay | 2 | 2 | 2 | 4 |
| FT-L2 | | | | |
| regularization coefficient | $10^4$ | $10^0$ | $10^2$ | $10^2$ |
| EWC | | | | |
| regularization coefficient | $10^{-2}$ | $10^0$ | $10^{-2}$ | $10^0$ |
| CSP | | | | |
| threshold | 0.1 | 0.1 | 0.1 | 0.1 |
| combination rollout length | 100 | 20 | 20 | 100 |

| Hyper-parameter | Ant | | | |
|---|---|---|---|---|
| | Forgetting | Transfer | Distraction | Compositional |
| lr policy | 0.001 | 0.001 | 0.001 | 0.0003 |
| lr critic | 0.001 | 0.001 | 0.001 | 0.0003 |
| reward scaling | 10. | 1. | 1. | 10. |
| target output std | 0.05 | 0.05 | 0.1 | 0.1 |
| policy update delay | 2 | 2 | 2 | 4 |
| target update delay | 4 | 2 | 4 | 4 |
| FT-L2 | | | | |
| regularization coefficient | $10^4$ | $10^0$ | $10^0$ | $10^2$ |
| EWC | | | | |
| regularization coefficient | $10^{-2}$ | $10^4$ | $10^2$ | $10^{-2}$ |
| CSP | | | | |
| threshold | 0.1 | 0.1 | 0.1 | 0.1 |
| combination rollout length | 100 | 100 | 100 | 20 |

| Hyper-parameter | Humanoid |
|---|---|
| lr policy | 0.001 |
| lr critic | 0.0003 |
| reward scaling | 0.1 |
| target output std | 0.1 |
| policy update delay | 1 |
| target update delay | 1 |
| FT-L2 | |
| regularization coefficient | $10^{-2}$ |
| EWC | |
| regularization coefficient | $10^{-2}$ |
| CSP | |
| threshold | 0.1 |
| combination rollout length | 100 |

## B  ENVIRONMENT DETAILS

### B.1  DESIGNING THE TASKS

We used the flexibility of Brax physics engine  (Freeman et al., 2021), and three of its continuous control environments **Halfcheetah** (obs dim: 18, action dim: 6), **Ant** (obs dim: 27, action dim: 8), and **Humanoid** (obs dim: 376, action dim: 17) to derive multiple tasks from them. To do so, we tweaked multiple environment parameters (Table  6)) and tried to learn a policy on these new tasks. From that pool of trials, we kept tasks (see Table  7) that were both challenging and diversified.

Table 6: Environment parameters tweaked to create interesting tasks for our scenarios. The Range column indicates the range of the multiplying factor applied to the environment parameter in question.

| Environment Parameter | Range | Description |
|---|---|---|
| mass | $[0.5, 1.5]$ | mass of a particular part of the agent's body (torso, legs, feet,...) |
| radius | $[0.5, 1.5]$ | radius of a particular part of the agent's body (torso, legs, feet,...) |
| gravity | $[0.15, 1.5]$ | gravity of the environment |
| friction | $[0.4, 1.5]$ | friction of the environment |
| actions | $\{1, -1\}$ | invert action values if set to $-1$. Used for distraction tasks. |
| observations (mask) | $[0.1, 0.8]$ | proportion of masked observations to simulate defective sensors |
| actions (mask) | $[0.25, 0.75]$ | proportion of masked actions to simulate defective modules |

Table 7: List of the 25 tasks used to create our scenarios and the parameter changes associated.

| | Task Name | Parameter changes |
|---|---|---|
| **HalfCheetah** | normal | {} |
| | carrystuff | {torso_mass: 4, torso_radius: 4} |
| | carrystuff_hugegravity | {torso_mass: 4, torso_radius: 4, gravity: 1.5} |
| | defectivesensor | {masked_obs: 0.5} |
| | hugefeet | {feet_mass: 1.5, feet_radius: 1.5} |
| | hugefeet_rainfall | {feet_mass: 1.5, feet_radius: 1.5, friction: 0.4} |
| | inverted_actions | {action_coefficient: -1.} |
| | moon | {gravity: 0.15} |
| | tinyfeet | {feet_mass: 0.5, feet_radius: 0.5} |
| | tinyfeet_moon | {feet_mass: 0.5, feet_radius: 0.5, gravity: 0.15} |
| | rainfall | {friction: 0.4} |
| **Ant** | normal | {} |
| | nofeet_2_3_4 | {action_mask: 0.75 (only 1st leg available)} |
| | nofeet_1_3_4 | {action_mask: 0.75 (only 2nd leg available)} |
| | nofeet_1_3 | {action_mask: 0.5 (1st diagonal diabled)} |
| | nofeet_2_4 | {action_mask: 0.5 (2nd diagonal diabled)} |
| | nofeet_1_2 | {action_mask: 0.5 (forefeet disabled)} |
| | nofeet_3_4 | {action_mask: 0.5 (hindfeet disabled)} |
| | inverted_actions | {action_coefficient: -1.} |
| | moon | {gravity: 0.7} |
| | rainfall | {friction: 0.4} |
| **Humanoid** | normal | {} |
| | moon | {gravity: 0.15} |
| | carrystuff | {torso_mass: 4, torso_radius: 4, lwaist_mass: 4, lwaist_radius: 4} |
| | tinyfeet | {shin_mass: 0.5, shin_radius: 0.5} |

### B.2 DESIGNING THE CONTINUAL RL SCENARIOS

Inspired by Wołczyk et al. (2021), we studied the relationship between these changes with a simple protocol: we learn a new task with a policy that has been pre-trained on a former task. We drew forgetting and transfer tables for each pair of tasks (see Figure 5). With this information, we designed 5 types of scenarios representing a particular challenge in continual learning. Note that all tasks have a budget of $1M$ interactions each and repeat their loop 2 times (i.e. $8M$ interactions in total) except for the Humanoid scenario that contains 4 tasks with $2M$ interactions.

1. **Forgetting Scenarios** are designed such that a single policy tends to forget the former task when learning a new one.
   - Halfcheetah: `hugefeet` → `moon` → `carrystuff` → `rainfall`
   - Ant: `normal` → `hugefeet` → `rainfall` → `moon`

2. **Transfer Scenarios** are designed such that a single policy has more difficulties to learn a new task after having learned the former one, rather than learning it from scratch.
   - Halfcheetah: `carrystuff_hugegravity` → `moon` → `defectivesensor` → `hugefeet_rainfall`
   - Ant: `nofeet_1_3` → `nofeet_2_4` → `nofeet_1_2` → `nofeet_3_4`

3. **Robustness Scenarios** alternate between a normal task and a very different distraction task that disturbs the whole learning process of a single policy (we simply inverted the actions). While this challenge looks particularly simple from a human perspective (a simple -1 vector applied on the output is fine to find an optimal policy in a continual setting), we figured out that the Fine-tuning policies struggle to recover good performances (the final average reward actually decreases).
   - Halfcheetah: `normal` → `inverted_actions` → `normal` → `inverted_actions`
   - Ant: `normal` → `inverted_actions` → `normal` → `inverted_actions`

4. **Compositional Scenarios** present two first tasks that will be useful to learn the last one, but a very different distraction task is put at the third place to disturb this forward transfer. The last task is indeed a combination of the two first tasks in the sense that it combines their particularities. For example, if the first task is "moon" and the second one is tinyfeet, the last one will combine moon's gravity and feet morphological changes.
   - Halfcheetah: `tinyfeet` → `moon` → `carrystuff_hugegravity` → `tinyfeet_moon`
   - Ant: `nofeet_2_3_4` → `nofeet_1_3_4` → `nofeet_1_2` → `nofeet_3_4`

5. **Humanoid scenario** is an additional scenario built with the challenging environment Humanoid to test our method in higher dimensions. Here is the detailed sequence of the scenario.
   - Humanoid: `normal` → `moon` → `carrystuff` → `tinyfeet`

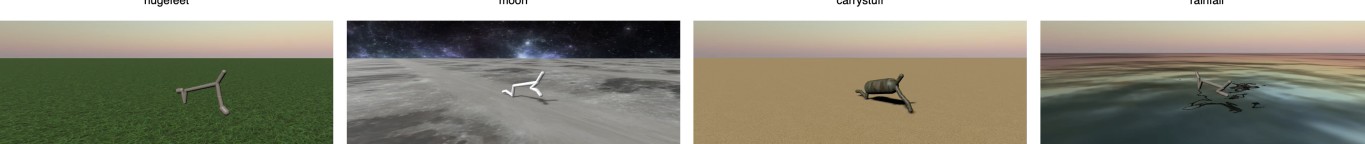

Figure 4: Rendering example of our Halfcheetah forgetting scenario. Click here for interactive visualizations.

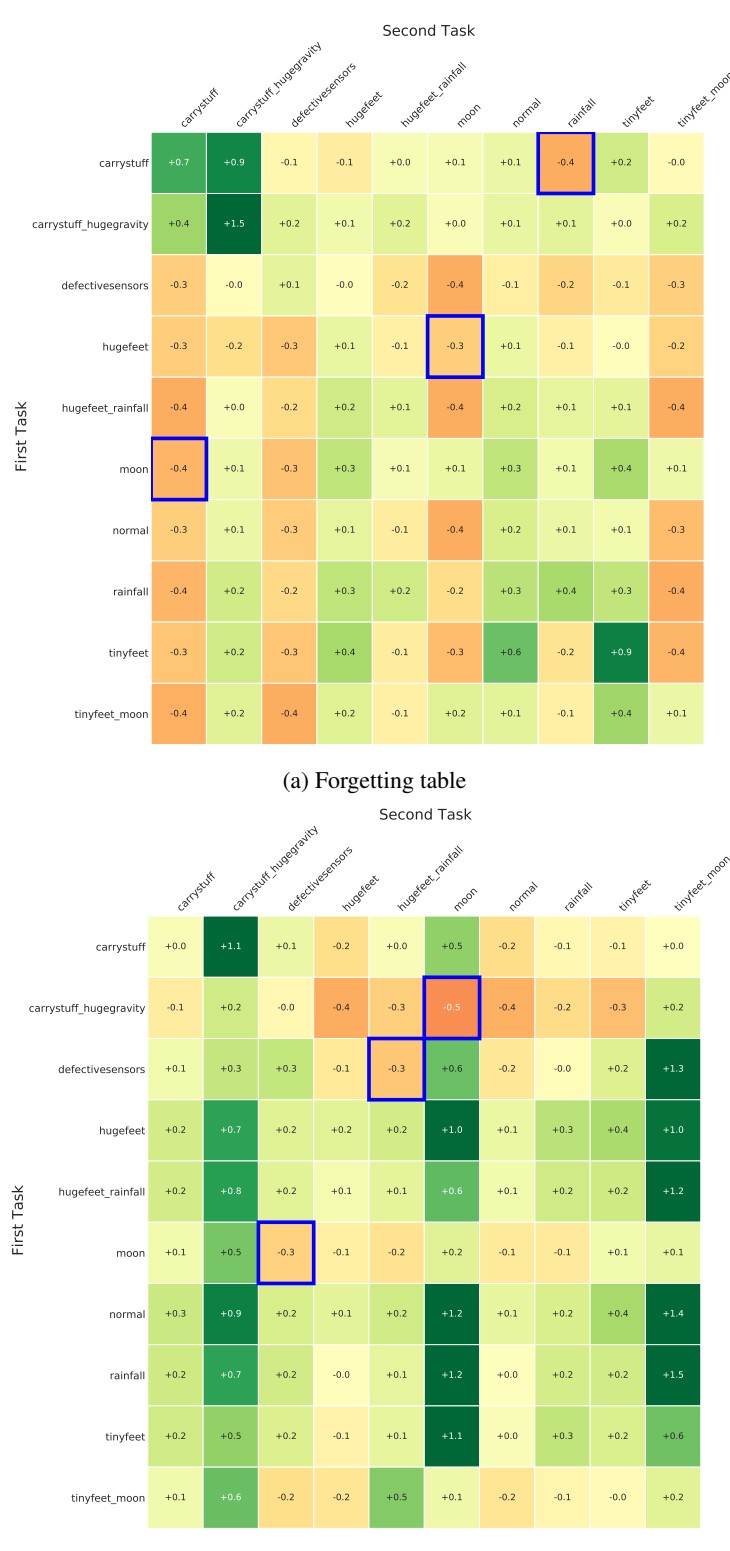

(a) Forgetting table

(b) Transfer table

Figure 5: Forgetting (a) and Transfer (b) tables of our Halfcheetah tasks. The pairs selected to create our scenario are highlighted in blue. Results are averaged over 3 seeds using a classical RL algorithm.

### B.3 CONTINUAL WORLD TASKS

The main benchmark of ContinualWorld is CW10, i.e. a sequence of 10 tasks designed by Wołczyk et al. (2021) such that it has an average transfer/forgetting difficulty. We also tested our model and our baselines on the 8 triplets (3-tasks scenarios) proposed by the authors. The tasks originally come from Metaworld (Yu et al., 2019), and have a budget of 1M interactions each. The scenarios were specially designed such that there is a positive forward transfer from task 1 to task 3, but task 2 is used as a distraction that interferes with these learning dynamics. These tasks are designed with the 2.0 version of the open-source MuJoCo physics engine Todorov et al. (2012). Here is the detailed sequence of the triplets:

1. `push-v1` → `window-close-v1` → `hammer-v1`

2. `hammer-v1` → `window-close-v1` → `faucet-close-v1`

3. `window-close-v1` → `handle-press-side-v1` → `peg-unplug-side-v1`

4. `faucet-close-v1` → `shelf-place-v1` → `peg-unplug-side-v1`

5. `faucet-close-v1` → `shelf-place-v1` → `push-back-v1`

6. `stick-pull-v1` → `peg-unplug-side-v1` → `stick-pull-v1`

7. `stick-pull-v1` → `push-back-v1` → `push-wall-v1`

8. `push-wall-v1` → `shelf-place-v1` → `push-back-v1`

## C DETAILS ABOUT CSP

### C.1 DETAILED ALGORITHM

---

**Algorithm 1:** Continual Subspace of Policies (CSP)

---

**Input:** $\theta_1, \ldots, \theta_j$ (previous anchors), $\epsilon$ (threshold)
**Initialize:** $W_\phi$ (subspace critic), $\mathcal{B}$ (replay buffer)
**Initialize:** $\theta_{j+1} \leftarrow \frac{1}{j} \sum_{i=1}^{j} \theta_i$ (new anchor) ;           // Grow the Subspace

1 **for** $i = 1, ..., B$ **do**
2     Sample $\alpha \sim Dir\big(\mathcal{U}(j+1)\big)$
3     Set policy parameters $\theta_\alpha \leftarrow \sum_{i=1}^{j+1} \alpha_i \theta_i$
4     **for** $l = 1, ..., K$ **do**
5         Collect and store $(s, a, r, s', \alpha)$ in $\mathcal{B}$ by sampling $a \sim \pi_{\theta_\alpha}(\cdot|s)$
6     **end**
7     **if** *time to update* **then**
8         Update $\pi_{\theta_{j+1}}$ and $W_\phi$ using the SAC algorithm and the replay buffer $\mathcal{B}$
9     **end**
10 **end**
11 Use $\mathcal{B}$ and $W_\phi$ to estimate: ;           // Extend or Prune the Subspace

$$\alpha^{\text{old}} \leftarrow \underset{(\alpha,0) \text{ with } \alpha \in \mathbb{R}^m_+, \|\alpha\|_1 = 1}{\arg\max} W_\phi(\alpha)$$

$$\alpha^{\text{new}} \leftarrow \underset{\alpha \in \mathbb{R}^{m+1}_+, \|\alpha\|_1 = 1}{\arg\max} W_\phi(\alpha)$$

   **if** $W_\phi(\cdot, \alpha^{\text{new}}) > (1 + \epsilon) \cdot W_\phi(\cdot, \alpha^{\text{old}})$ **then**
13     **Return:** $\theta_1, \ldots, \theta_j, \theta_{j+1}, \alpha^{\text{new}}$;           // Extend
14 **else**
15     **Return:** $\theta_1, \ldots, \theta_j, \alpha^{\text{old}}$;           // Prune
16 **end**

---

## C.2 SCALABILITY

By having access to an infinite number of policies, the subspace is highly expressive so it can capture a wide range of diverse behaviors. This enables positive transfer to many new tasks without the need for training additional parameters. As a consequence, the number of parameters scales *sublinearly* with the number of tasks. The speed of growth is controlled by the threshold $\epsilon$ which defines how much performance we are willing to give up for decreasing the number of parameters (by the size of one policy network). Practitioners can set the threshold to trade-off performance gains for memory efficiency (*i.e.* the higher the $\epsilon$ the more performance losses are tolerated to reduce memory costs). In practice, we noticed that setting $\epsilon = 0.1$ allows good performance and a limited growth of parameters.

As the agent learns to solve more and more tasks, we expect the subspace to grow more slowly (or stop growing entirely) since it already contains many useful behaviors which *transfer* to new tasks. On the other hand, if the agent encounters a task that is significantly different than previous ones and all other behaviors in the subspace, the subspace still has the *flexibility* to grow and incorporate entirely new skills. The number of anchors in the final subspace is adaptive and depends on the sequence of tasks. The longer and more diverse the task sequence, the more anchors are needed. This property is important for real-world applications with open-ended interactions where it's unlikely to know a priori how much capacity is required to express all useful skills.

## C.3 TRAINING AND USING THE CRITIC

The subspace critic $W_\phi$ plays a central role in our method. Compared to the vanilla SAC critic, we only add the convex combination $\alpha$ as an input (concatenated with the states and actions). In this way, it is optimized not only to evaluate the future averaged return on $(s, a)$ pairs of a single policy, but an infinity of policies, characterized by the convex combination $\alpha$.

At the end of each task, the subspace critic has the difficult task to estimate by how much the new anchor policy $\theta_{j+1}$ improves the performance of the current subspace $\Theta_j$. To do so, one has to find the best combination of policies in the last subspace $\Theta_j$, calling it $\alpha^{old}$ and the one in the current subspace $\Theta_{j+1}$, calling it $\alpha^{new}$. In practice, we found that sampling 1024 random $(s, a)$ pairs from the replay buffer at the end of the task allows to have an accurate estimation of the best policies. This part does not require any new interaction with the environment.

However, we found that rolling out the top-k $\alpha$ in the new and former subspaces helps to find the best combination. In practice, we found that setting $k := 8$ with one rollout per combination is sufficient. In tasks that have $1M$ interactions and an episode horizon of 1000, it requires to allocate $0.8\%$ of the budget to the purpose of finding a good policy in the subspace, which does not significantly impact the training procedure.

## C.4 SAMPLING POLICIES FROM THE SUBSPACE

Yet, it is important to allow the critic to estimate $\alpha^{old}$. During training, we noticed that sampling with a simple flat Dirichlet distribution (i.e. a uniform distribution over the simplex induced by the current subspace) is not enough to make the critic able to accurately estimate the performance of the last subspace (indeed, the chances of sampling a policy in the last subspace are almost surely 0.). This is why we decided to sample in both the current and the last subspace. The distribution we use is then a mixture of two Dirichlet (equal chances of sampling in the last subspace and in the current subspace). We did not perform an ablation to see if balancing the mixture would increase performances.

We also tried to sample with a peaked distribution (concentration of the Dirichlet equal to the inverse of the number of the anchors) to see if it increased performances. In some cases the new subspace is able to find good policies faster with this distribution. It can be a good trade off between always choosing the last anchor and sampling uniformly.

## C.5 IMPLEMENTATION

Our Pytorch implementation of CSP uses the `nn.ModuleList` object to store anchor networks. The additional computational cost compared to a single network is negligible during both training and inference as it is mentioned in Wortsman et al. (2021).

### C.6 ANALYSIS OF THE SUBSPACE

The he best way to visualize the reward and critic value landscapes of the subspaces is when there are 3 anchors (see Figure 3). To do saw, we draw 8192 evenly spaced points in the 3-dimensional simplex of $\mathbb{R}^3$, and average the return over 10 rollouts for the reward landscape, and 1024 pairs of $(s, a)$ for the critic landscape. We used the short version of the Compositional Scenario of HalfCheetah to display the results.

### C.7 LIMITATIONS

CSP prevents forgetting of prior tasks, promotes transfer to new tasks, and scales sublinearly with the number of tasks. Despite all these advantages, our method still has a number of limitations. While the subspace grows only sublinearly with the number of tasks, this number is highly dependent on the task sequence. In the worst case scenario, it increases linearly with the number of tasks. On the other hand, the more similar the tasks are, the lower the size of the subspace needed to learn good policies for all tasks. Thus, one important direction for future work is to learn a subspace of policies with a fixed number of anchors. Instead of training an additional anchor for each new task, one could optimize a policy in the current subspace on the new task while ensuring that the best policies for prior tasks don't change too much. This could be formulated as a constrained optimization problem where all the anchors defining the subspace are updated for each new task, but some regions of the subspace are regularized to not change very much. This would result in the subspace having different regions which are good for different tasks.

While the memory costs increase sublinearly with the number of tasks, the computational costs increase linearly with the number of tasks, in the current implementation of CSP. This is because we train an additional anchor for each new task, which can be removed if it doesn't significantly improve performance. However, the computational costs can also be reduced if you have access to maximum reward on a task. This is typically the case for sparse reward tasks where if the agent succeeds, it receives a reward of 1 and 0 otherwise. In this case, there is no need to train one anchor per task. Instead, one can simply find the best policy in the current subspace and compare its performance with the maximum reward to decide whether to train an additional anchor or not. Hence, in this version of CSP (which is a minor modification of the current implementation) both memory and compute scale sublinearly with the number of tasks. However, this assumption doesn't always hold, so here we decided to implement the more general version of CSP.

In this work, we don't specifically leverage the structure of the subspace in order to find good policies. Hence, one promising research direction is to further improve transfer efficiency by leveraging the structure of the subspace to find good policies for new tasks. For example, this could be done by finding the convex combination of the anchors which maximizes return on a given task. Regularizing the geometry of the subspace to impose certain inductive biases could also be a fruitful direction for future work.

# D    ADDITIONAL RESULTS

## D.1    BRAX

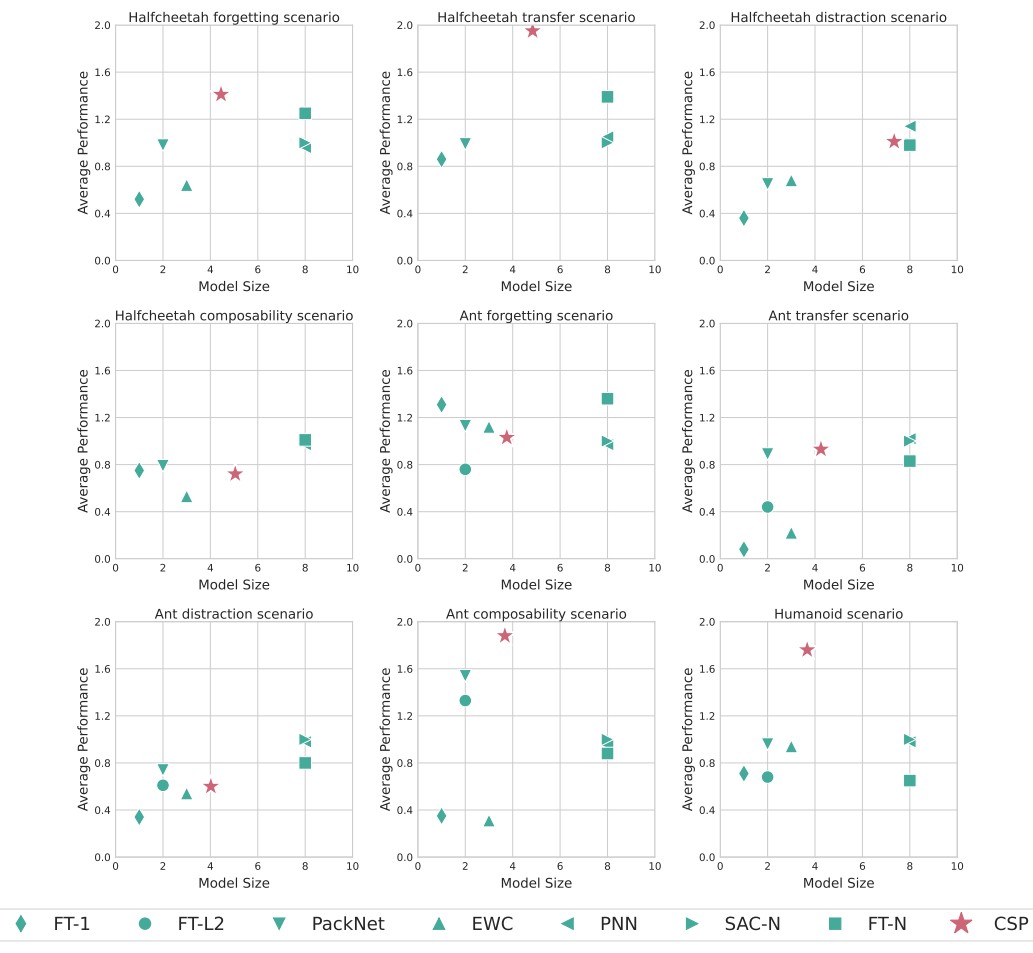

Figure 6: Performance-size trade-off for the 9 continuous control scenarios from Brax. Results are averaged over 10 seeds.

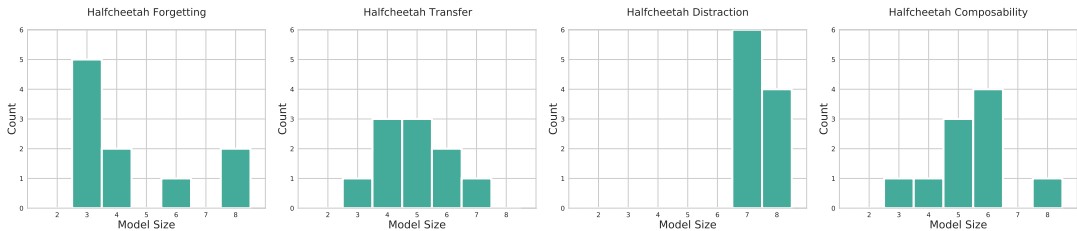

Figure 7: Histogram of the Model Size of CSP when trained on HalfCheetah scenarios (10 seeds per scenario).

Table 8: Detailed results of the long version (8 tasks) of our 4 Halfcheetah scenarios. Results presents mean and standard deviation of our 4 metrics, are split by scenario and method, and are averaged over 10 seeds.

| | Method | Performance | Model size | Transfer | Forgetting |
|---|---|---|---|---|---|
| **Forgetting scenario** | FT-1 | $0.52 \pm 0.08$ | $\mathbf{1.0 \pm 0.0}$ | $0.19 \pm 0.23$ | $0.67 \pm 0.19$ |
| | FT-L2 | $0.67 \pm 0.32$ | $2.0 \pm 0.0$ | $-0.34 \pm 0.3$ | $-0.01 \pm 0.0$ |
| | PACKNET | $0.94 \pm 0.18$ | $2.0 \pm 0.0$ | $-0.07 \pm 0.17$ | $-0.0 \pm 0.0$ |
| | PACKNETX2 | $1.00 \pm 0.05$ | $4.0 \pm 0.0$ | $0.00 \pm 0.05$ | $-0.0 \pm 0.0$ |
| | EWC | $0.64 \pm 0.26$ | $3.0 \pm 0.0$ | $-0.27 \pm 0.31$ | $0.09 \pm 0.13$ |
| | PNN | $0.96 \pm 0.15$ | $8.0 \pm 0.0$ | $-0.04 \pm 0.13$ | $0.0 \pm 0.0$ |
| | SAC-N | $1.0 \pm 0.1$ | $8.0 \pm 0.0$ | $-0.0 \pm 0.09$ | $-0.0 \pm 0.0$ |
| | FT-N | $1.25 \pm 0.24$ | $8.0 \pm 0.0$ | $0.25 \pm 0.23$ | $0.0 \pm 0.0$ |
| | CSP-ORACLE | $1.60 \pm 0.12$ | $4.5 \pm 2.0$ | $0.38 \pm 0.08$ | $-0.22 \pm 0.15$ |
| | CSP | $\mathbf{1.41 \pm 0.07}$ | | $\mathbf{0.41 \pm 0.06}$ | $\mathbf{0.00 \pm 0.0}$ |
| **Transfer scenario** | FT-1 | $0.86 \pm 0.7$ | $\mathbf{1.0 \pm 0.0}$ | $0.52 \pm 0.62$ | $0.66 \pm 0.42$ |
| | FT-L2 | $-0.03 \pm 0.07$ | $2.0 \pm 0.0$ | $-1.00 \pm 0.03$ | $-0.03 \pm 0.04$ |
| | PACKNET | $0.99 \pm 0.25$ | $2.0 \pm 0.0$ | $-0.01 \pm 0.24$ | $0.0 \pm 0.0$ |
| | PACKNETX2 | $1.03 \pm 0.12$ | $4.0 \pm 0.0$ | $0.03 \pm 0.12$ | $0.0 \pm 0.0$ |
| | EWC | $-0.13 \pm 0.23$ | $3.0 \pm 0.0$ | $-1.13 \pm 0.21$ | $0.0 \pm 0.02$ |
| | PNN | $1.05 \pm 0.14$ | $8.0 \pm 0.0$ | $0.04 \pm 0.13$ | $-0.0 \pm 0.0$ |
| | SAC-N | $1.0 \pm 0.15$ | $8.0 \pm 0.0$ | $-0.0 \pm 0.14$ | $-0.0 \pm 0.0$ |
| | FT-N | $1.39 \pm 0.34$ | $8.0 \pm 0.0$ | $0.39 \pm 0.33$ | $0.0 \pm 0.01$ |
| | CSP-ORACLE | $3.75 \pm 0.49$ | $4.9 \pm 1.1$ | $2.09 \pm 0.71$ | $-0.66 \pm 0.6$ |
| | CSP | $\mathbf{1.95 \pm 0.83}$ | | $\mathbf{0.93 \pm 0.79}$ | $\mathbf{-0.01 \pm 0.03}$ |
| **Distraction scenario** | FT-1 | $0.36 \pm 0.25$ | $1.0 \pm 0.0$ | $-0.11 \pm 0.2$ | $0.53 \pm 0.25$ |
| | FT-L2 | $0.22 \pm 0.16$ | $2.0 \pm 0.0$ | $-0.79 \pm 0.15$ | $-0.0 \pm 0.0$ |
| | PACKNET | $0.65 \pm 0.11$ | $2.0 \pm 0.0$ | $-0.35 \pm 0.1$ | $0.0 \pm 0.0$ |
| | PACKNETX2 | $0.73 \pm 0.12$ | $4.0 \pm 0.0$ | $-0.27 \pm 0.12$ | $0.0 \pm 0.0$ |
| | EWC | $0.68 \pm 0.28$ | $3.0 \pm 0.0$ | $-0.31 \pm 0.23$ | $0.01 \pm 0.09$ |
| | PNN | $\mathbf{1.14 \pm 0.1}$ | $8.0 \pm 0.0$ | $\mathbf{0.14 \pm 0.1}$ | $0.0 \pm 0.0$ |
| | SAC-N | $1.0 \pm 0.29$ | $8.0 \pm 0.0$ | $0.0 \pm 0.28$ | $0.0 \pm 0.0$ |
| | FT-N | $0.98 \pm 0.12$ | $8.0 \pm 0.0$ | $-0.02 \pm 0.11$ | $-0.0 \pm 0.0$ |
| | CSP-ORACLE | $1.28 \pm 0.09$ | $7.4 \pm 0.5$ | $0.19 \pm 0.08$ | $-0.09 \pm 0.08$ |
| | CSP | $1.01 \pm 0.13$ | | $0.01 \pm 0.12$ | $\mathbf{-0.0 \pm 0.01}$ |
| **Composability scenario** | FT-1 | $0.75 \pm 0.12$ | $\mathbf{1.0 \pm 0.0}$ | $-0.04 \pm 0.09$ | $0.22 \pm 0.11$ |
| | FT-L2 | $0.66 \pm 0.03$ | | $-0.35 \pm 0.03$ | $0.01 \pm 0.03$ |
| | PACKNET | $0.79 \pm 0.03$ | $2.0 \pm 0.0$ | $-0.21 \pm 0.03$ | $-0.0 \pm 0.0$ |
| | PACKNETX2 | $0.88 \pm 0.08$ | $4.0 \pm 0.0$ | $-0.12 \pm 0.08$ | $0.0 \pm 0.0$ |
| | EWC | $0.53 \pm 0.17$ | $3.0 \pm 0.0$ | $-0.34 \pm 0.09$ | $0.13 \pm 0.12$ |
| | PNN | $0.97 \pm 0.16$ | $8.0 \pm 0.0$ | $-0.03 \pm 0.16$ | $0.0 \pm 0.0$ |
| | SAC-N | $1.0 \pm 0.05$ | $8.0 \pm 0.0$ | $-0.0 \pm 0.05$ | $-0.0 \pm 0.0$ |
| | FT-N | $\mathbf{1.01 \pm 0.09}$ | $8.0 \pm 0.0$ | $\mathbf{0.01 \pm 0.09}$ | $0.0 \pm 0.0$ |
| | CSP-ORACLE | $0.87 \pm 0.07$ | $3.4 \pm 1.5$ | $-0.1 \pm 0.07$ | $0.03 \pm 0.06$ |
| | CSP | $0.69 \pm 0.09$ | | $-0.31 \pm 0.09$ | $\mathbf{0.0 \pm 0.0}$ |
| **Aggregate** | FT-1 | $0.62 \pm 0.29$ | $\mathbf{1.0 \pm 0.0}$ | $0.14 \pm 0.29$ | $0.52 \pm 0.24$ |
| | FT-L2 | $0.38 \pm 0.15$ | $2.0 \pm 0.0$ | $-0.62 \pm 0.13$ | $-0.01 \pm 0.02$ |
| | PACKNET | $0.85 \pm 0.14$ | $2.0 \pm 0.0$ | $-0.15 \pm 0.09$ | $0.0 \pm 0.0$ |
| | PACKNETX2 | $0.91 \pm 0.09$ | $4.0 \pm 0.0$ | $-0.09 \pm 0.09$ | $0.0 \pm 0.0$ |
| | EWC | $0.43 \pm 0.24$ | $3.0 \pm 0.0$ | $-0.51 \pm 0.21$ | $0.06 \pm 0.09$ |
| | PNN | $1.03 \pm 0.14$ | $8.4 \pm 0.0$ | $0.03 \pm 0.13$ | $0.0 \pm 0.0$ |
| | SAC-N | $1.0 \pm 0.15$ | $8.0 \pm 0.0$ | $0.0 \pm 0.14$ | $0.0 \pm 0.0$ |
| | FT-N | $1.16 \pm 0.2$ | $8.0 \pm 0.0$ | $0.16 \pm 0.19$ | $0.0 \pm 0.0$ |
| | CSP-ORACLE | $1.88 \pm 0.19$ | $5.4 \pm 1.3$ | $0.64 \pm 0.24$ | $-0.24 \pm 0.22$ |
| | CSP | $\mathbf{1.27 \pm 0.27}$ | | $\mathbf{0.27 \pm 0.26}$ | $\mathbf{0.0 \pm 0.01}$ |

Table 9: Detailed results of the long version (8 tasks) of our 4 Ant scenarios. Results presents mean and standard deviation of our 4 metrics, are split by scenario and method, and are averaged over 10 seeds.

| | Method | Performance | Model size | Transfer | Forgetting |
|---|---|---|---|---|---|
| **Forgetting scenario** | FT-1 | $1.31 \pm 0.33$ | $\mathbf{1.0 \pm 0.0}$ | $0.36 \pm 0.2$ | $0.05 \pm 0.23$ |
| | FT-L2 | $0.76 \pm 0.27$ | $2.0 \pm 0.0$ | $-0.24 \pm 0.24$ | $0.0 \pm 0.04$ |
| | PACKNET | $1.13 \pm 0.2$ | $2.0 \pm 0.0$ | $0.13 \pm 0.19$ | $0.0 \pm 0.0$ |
| | EWC | $1.12 \pm 0.21$ | $3.0 \pm 0.0$ | $0.3 \pm 0.15$ | $0.17 \pm 0.22$ |
| | PNN | $0.97 \pm 0.2$ | $8.0 \pm 0.0$ | $-0.03 \pm 0.19$ | $0.0 \pm 0.0$ |
| | SAC-N | $1.0 \pm 0.17$ | $8.0 \pm 0.0$ | $-0.0 \pm 0.16$ | $0.0 \pm 0.0$ |
| | FT-N | $\mathbf{1.36 \pm 0.26}$ | $8.0 \pm 0.0$ | $\mathbf{0.36 \pm 0.25}$ | $-0.0 \pm 0.0$ |
| | CSP-ORACLE | $1.24 \pm 0.08$ | $3.7 \pm 1.2$ | $0.11 \pm 0.06$ | $-0.13 \pm 0.02$ |
| | CSP | $1.03 \pm 0.14$ | | $0.03 \pm 0.13$ | $\mathbf{0.0 \pm 0.0}$ |
| **Transfer scenario** | FT-1 | $0.08 \pm 0.14$ | $\mathbf{1.0 \pm 0.0}$ | $-0.28 \pm 0.2$ | $0.64 \pm 0.15$ |
| | FT-L2 | $0.44 \pm 0.12$ | $2.0 \pm 0.0$ | $-0.44 \pm 0.07$ | $0.12 \pm 0.09$ |
| | PACKNET | $0.89 \pm 0.09$ | $2.0 \pm 0.0$ | $-0.11 \pm 0.09$ | $-0.0 \pm 0.0$ |
| | EWC | $0.22 \pm 0.05$ | $3.0 \pm 0.0$ | $-0.78 \pm 0.04$ | $0.0 \pm 0.0$ |
| | PNN | $1.02 \pm 0.05$ | $8.0 \pm 0.0$ | $0.02 \pm 0.05$ | $0.0 \pm 0.0$ |
| | SAC-N | $\mathbf{1.0 \pm 0.08}$ | $8.0 \pm 0.0$ | $\mathbf{0.0 \pm 0.07}$ | $-0.0 \pm 0.0$ |
| | FT-N | $0.83 \pm 0.12$ | $8.0 \pm 0.0$ | $-0.17 \pm 0.12$ | $-0.0 \pm 0.0$ |
| | CSP-ORACLE | $1.02 \pm 0.09$ | $4.3 \pm 0.6$ | $0.0 \pm 0.07$ | $-0.02 \pm 0.02$ |
| | CSP | $0.93 \pm 0.1$ | | $-0.07 \pm 0.09$ | $\mathbf{-0.0 \pm 0.0}$ |
| **Distraction scenario** | FT-1 | $0.34 \pm 0.06$ | $\mathbf{1.0 \pm 0.0}$ | $-0.16 \pm 0.04$ | $0.5 \pm 0.09$ |
| | FT-L2 | $0.61 \pm 0.08$ | $2.0 \pm 0.0$ | $-0.42 \pm 0.05$ | $-0.03 \pm 0.06$ |
| | PACKNET | $0.74 \pm 0.05$ | $2.0 \pm 0.0$ | $-0.26 \pm 0.04$ | $0.0 \pm 0.0$ |
| | EWC | $0.54 \pm 0.08$ | $3.0 \pm 0.0$ | $-0.47 \pm 0.07$ | $-0.01 \pm 0.02$ |
| | PNN | $0.98 \pm 0.19$ | $8.0 \pm 0.0$ | $-0.02 \pm 0.18$ | $-0.0 \pm 0.0$ |
| | SAC-N | $\mathbf{1.0 \pm 0.09}$ | $8.0 \pm 0.0$ | $\mathbf{-0.0 \pm 0.09}$ | $-0.0 \pm 0.0$ |
| | FT-N | $0.8 \pm 0.09$ | $8.0 \pm 0.0$ | $-0.2 \pm 0.09$ | $-0.0 \pm 0.0$ |
| | CSP-ORACLE | $0.57 \pm 0.08$ | $4.0 \pm 0.8$ | $-0.35 \pm 0.1$ | $0.08 \pm 0.09$ |
| | CSP | $0.6 \pm 0.11$ | | $-0.4 \pm 0.1$ | $\mathbf{0.0 \pm 0.0}$ |
| **Composability scenario** | FT-1 | $0.35 \pm 0.49$ | $\mathbf{1.0 \pm 0.0}$ | $0.32 \pm 0.89$ | $0.97 \pm 0.73$ |
| | FT-L2 | $1.33 \pm 0.35$ | $2.0 \pm 0.0$ | $0.08 \pm 0.37$ | $-0.25 \pm 0.18$ |
| | PACKNET | $1.54 \pm 0.5$ | $2.0 \pm 0.0$ | $0.54 \pm 0.47$ | $-0.0 \pm 0.0$ |
| | EWC | $0.31 \pm 0.62$ | $3.0 \pm 0.0$ | $-0.07 \pm 0.47$ | $0.62 \pm 0.27$ |
| | PNN | $0.95 \pm 0.81$ | $8.0 \pm 0.0$ | $-0.05 \pm 0.77$ | $-0.0 \pm 0.0$ |
| | SAC-N | $1.0 \pm 1.17$ | $8.0 \pm 0.0$ | $0.0 \pm 1.11$ | $0.0 \pm 0.0$ |
| | FT-N | $0.88 \pm 0.35$ | $8.0 \pm 0.0$ | $-0.12 \pm 0.34$ | $-0.0 \pm 0.0$ |
| | CSP-ORACLE | $2.15 \pm 0.02$ | $3.6 \pm 0.4$ | $1.04 \pm 0.23$ | $-0.11 \pm 0.24$ |
| | CSP | $\mathbf{1.88 \pm 0.33}$ | | $\mathbf{0.88 \pm 0.32}$ | $\mathbf{-0.0 \pm 0.01}$ |
| **Aggregate** | FT-1 | $0.52 \pm 0.26$ | $\mathbf{1.0 \pm 0.0}$ | $0.06 \pm 0.33$ | $0.54 \pm 0.3$ |
| | FT-L2 | $0.78 \pm 0.2$ | $2.0 \pm 0.0$ | $-0.25 \pm 0.18$ | $-0.04 \pm 0.09$ |
| | PACKNET | $1.08 \pm 0.21$ | $2.0 \pm 0.0$ | $0.08 \pm 0.2$ | $0.0 \pm 0.0$ |
| | EWC | $0.55 \pm 0.24$ | $3.0 \pm 0.0$ | $-0.26 \pm 0.18$ | $0.2 \pm 0.13$ |
| | PNN | $0.98 \pm 0.31$ | $8.0 \pm 0.0$ | $-0.02 \pm 0.3$ | $0.0 \pm 0.0$ |
| | SAC-N | $1.0 \pm 0.38$ | $8.0 \pm 0.0$ | $0.0 \pm 0.36$ | $0.0 \pm 0.0$ |
| | FT-N | $0.97 \pm 0.2$ | $8.0 \pm 0.0$ | $-0.03 \pm 0.2$ | $0.0 \pm 0.0$ |
| | CSP-ORACLE | $1.24 \pm 0.07$ | $3.9 \pm 0.8$ | $0.20 \pm 0.06$ | $-0.05 \pm 0.16$ |
| | CSP | $\mathbf{1.11 \pm 0.17}$ | | $\mathbf{0.11 \pm 0.16}$ | $\mathbf{0.0 \pm 0.0}$ |

Table 10: Detailed results of our Humanoid scenario. Results presents mean and standard deviation of our 4 metrics, are split by method, and are averaged over 10 seeds.

| Method | Performance | Model size | Transfer | Forgetting |
|---|---|---|---|---|
| FT-1 | $0.71 \pm 0.07$ | $1.0 \pm 0.0$ | $0.1 \pm 0.23$ | $0.38 \pm 0.27$ |
| FT-L2 | $0.68 \pm 0.28$ | $2.0 \pm 0.0$ | $0.01 \pm 0.31$ | $0.33 \pm 0.28$ |
| PACKNET | $0.96 \pm 0.21$ | $2.0 \pm 0.0$ | $-0.04 \pm 0.2$ | $-0.0 \pm 0.0$ |
| EWC | $0.94 \pm 0.01$ | $3.0 \pm 0.0$ | $-0.05 \pm 0.02$ | $0.01 \pm 0.02$ |
| PNN | $0.98 \pm 0.26$ | $4.0 \pm 0.0$ | $-0.02 \pm 0.3$ | $0.0 \pm 0.0$ |
| SAC-N | $1.0 \pm 0.29$ | $4.0 \pm 0.0$ | $0.0 \pm 0.21$ | $-0.0 \pm 0.0$ |
| FT-N | $0.65 \pm 0.46$ | $4.0 \pm 0.0$ | $-0.35 \pm 0.35$ | $-0.0 \pm 0.0$ |
| CSP-ORACLE | $1.98 \pm 0.22$ | $3.4 \pm 0.3$ | $0.93 \pm 0.12$ | $-0.05 \pm 0.32$ |
| CSP | $1.76 \pm 0.19$ | | $0.75 \pm 0.16$ | $-0.0 \pm 0.0$ |

Table 11: Raw cumulative rewards obtained on each task of the 4 scenarios. These are obtained with a single policy learned from scratch (SAC-N) and the set of hyperparameters selected by the gridsearch of the FT-N method (see A). This explains why similar tasks have different averaged returns across scenarios. Results are averaged over 3 seeds.

| | Scenario | Task | Cumulative rewards |
|---|---|---|---|
| **Halfcheetah** | Forgetting Scenario | hugefoot | 2209 |
| | | moon | 2982 |
| | | carrystuff | 6309 |
| | | rainfall | 1001 |
| | Transfer Scenario | carrystuff_hugegravity | 7233 |
| | | moon | 3599 |
| | | defectivemodule | 5909 |
| | | hugefoot_rainfall | 2942 |
| | Distraction Scenario | normal | 4932 |
| | | inverted_actions | 5833 |
| | | normal | 4932 |
| | | inverted_actions | 5833 |
| | Compositional Scenario | tinyfoot | 6311 |
| | | moon | 3932 |
| | | carrystuff_hugegravity | 6319 |
| | | tinyfoot_moon | 1355 |
| **Ant** | Forgetting Scenario | normal | 3752 |
| | | hugefeet | 2841 |
| | | rainfall | 1596 |
| | | moon | 1401 |
| | Transfer Scenario | disableddiagonalfeet_1 | 3021 |
| | | disableddiagonalfeet_2 | 4119 |
| | | disabledforefeet | 1014 |
| | | disabledhindfeet | 1021 |
| | Distraction Scenario | normal | 3542 |
| | | inverted_actions | 4199 |
| | | normal | 3542 |
| | | inverted_actions | 4199 |
| | Compositional Scenario | disabled3feet_1 | 770 |
| | | disabled3feet_2 | 641 |
| | | disabledforefeet | 201 |
| | | disabledhindfeet | 288 |
| **Humanoid** | | normal | 1958 |
| | | moon | 1691 |
| | | carrystuff | 2379 |
| | | tinyfeet | 1711 |

### D.2 CONTINUAL WORLD

In addition to the CW10 benchmark, we also tried our method over the 8 triplets (3 tasks for each scenario) proposed in Wołczyk et al. (2021). In these experiments, we used CSP on the whole network. To that end, we did not add the layer normalization described in Appendix A.2, since it is not trivial to design a subspace in these sets of parameters. See Table 12 for the results.

Table 12: Detailed results of 8 triplets scenarios from Continual World. Results presents mean and standard deviation of the final average performance, are split by scenario and method, and are averaged over 3 seeds.

| Method | T1 | T2 | T3 | T4 | T5 | T6 | T7 | T8 | Agregate |
|---|---|---|---|---|---|---|---|---|---|
| FT-1 | 0.24 ± 0.13 | 0.25 ± 0.07 | 0.39 ± 0.16 | 0.34 ± 0.05 | 0.30 ± 0.01 | 0.32 ± 0.25 | 0.17 ± 0.07 | 0.34 ± 0.05 | 0.29 ± 0.1 |
| EWC | 0.45 ± 0.12 | 0.27 ± 0.09 | 0.38 ± 0.09 | 0.31 ± 0.12 | 0.32 ± 0.07 | 0.33 ± 0.18 | 0.20 ± 0.10 | 0.32 ± 0.08 | 0.32 ± 0.11 |
| PNN | **0.84 ± 0.08** | 0.72 ± 0.17 | 0.90 ± 0.05 | 0.43 ± 0.08 | 0.33 ± 0.23 | 0.46 ± 0.21 | 0.44 ± 0.12 | 0.36 ± 0.2 | 0.56 ± 0.14 |
| SAC-N | 0.69 ± 0.17 | 0.71 ± 0.13 | 0.79 ± 0.19 | 0.47 ± 0.14 | **0.60 ± 0.13** | 0.55 ± 0.11 | 0.54 ± 0.15 | 0.45 ± 0.12 | 0.6 ± 0.14 |
| FT-N | 0.77 ± 0.08 | **0.86 ± 0.1** | 0.78 ± 0.15 | 0.49 ± 0.14 | 0.52 ± 0.13 | **0.61 ± 0.06** | **0.61 ± 0.13** | 0.52 ± 0.06 | 0.65 ± 0.11 |
| CSP (ours) | 0.76 ± 0.2 | 0.79 ± 0.03 | **0.82 ± 0.08** | **0.58 ± 0.09** | 0.58 ± 0.06 | 0.54 ± 0.06 | 0.58 ± 0.04 | **0.53 ± 0.08** | **0.65 ± 0.08** |

We conducted a qualitative analysis to determine whether forward transfer also comes from the compositionality induced by the subspace (just like in HalfCheetah domain, see Figure 3b). To do so, we used the 3 first anchors created by CSP on the three first tasks of CW10, and computed the reward landscape of each of the 10 tasks. See Figure 8. We note that :

- **Smoothness:** As for Brax scenarios, the reward landscape remains smooth on all the figures.
- **Diversity:** Policies in the subspace specialize in some tasks that might share similarities. while `Push-Back` (d) is tackled by almost all policies of the subspaces, we see that policies on the lower left are good at tackling `Push-Wall` (b), `Faucet-Close` (b) and `Peg-Unplug-Side` (j). On the right side of the subspace, policies tackle `Window-Close` (i). On the top, policies are good at resolving `Hammer` (a) and `Handle-Press-Side` (f).
- **Compositionality:** the subspace show great compositionality on `Window-Close` (i) in a space around anchors instantiated on `Hammer` and `Faucet-Close` (some points reach 100% success rate directly).

In addition, while `Shelf-Place` (h) show zero transfer (it is the hardest task according to the learning curves in Wołczyk et al. (2021), all others show different degrees of forward transfer. Interestingly, `Push-Back` (d) is also tackled by almost all policies of the subspaces, while it is not the hardest task according to the learning curves Wołczyk et al. (2021).

### D.3 ABLATIONS

Ablations on the threshold parameter (see Figure 2) of CSP have been performed on the Robustness scenario of HalfCheetah. Concerning the scalability ablation, we used the Compositional scenario of HalfCheetah and duplicated it in 3 versions (4 tasks, 8 tasks, 12 tasks). Concerning the learning efficiency, we used the Robustness Scenario (short version, i.e. 4 tasks only) and ran it with a budget of 250k,500k and 1M interactions. Results were averaged over 3 seeds.

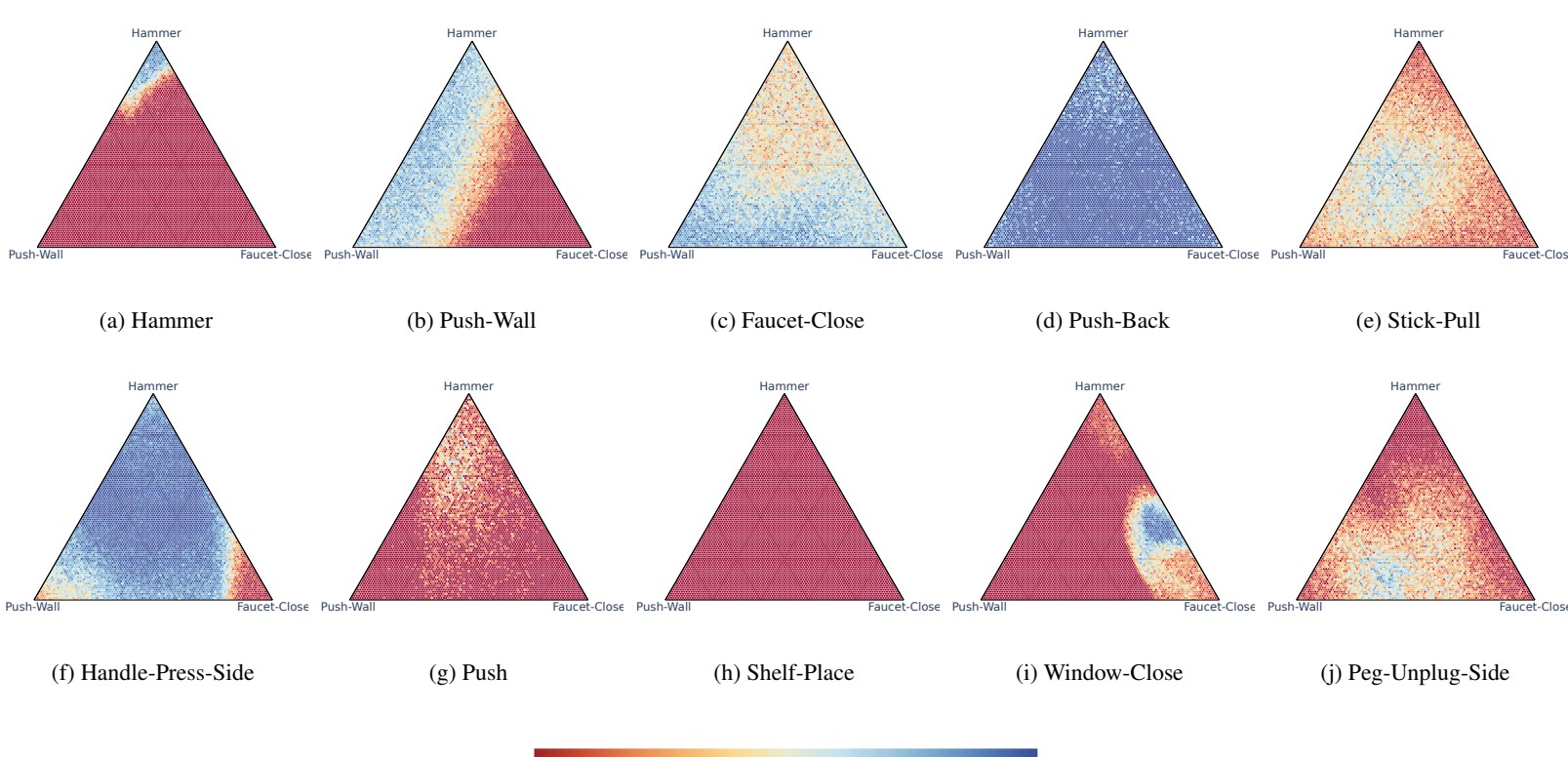

Figure 8: Reward landscape on each CW10 task when taking policies between the 3 first anchors (learned on `Hammer`, `Push-Wall` and `Faucet-Close`). Each of the 5151 points represent a policy. The results are averaged over 20 rollouts (we take the deterministic form of the policy to infer the trajectory).

