# OpenReview forum: "Building a Subspace of Policies for Scalable Continual Learning"
_ICLR.cc/2023/Conference — ICLR 2023 notable top 25%_

### Official Review · Reviewer_9jow · 2022-10-22

**Confidence:** 2
**Correctness:** 3
**Technical Novelty And Significance:** 2
**Empirical Novelty And Significance:** 2
**Recommendation:** 8

**Clarity, Quality, Novelty And Reproducibility:**

Paper is clear. Experimental results very detailed. Good explanation of the limitations.

**Strength And Weaknesses:**

Strength:
- The paper is well written, very detailed and thorough in its evaluation.
- The problem setting is sound and very relevant to lifelong learning.
- Very good experimental analysis of the transfer / forgetting properties of the approach.
- The fact that performance on the benchmarks that the method is evaluated on is close to oracle performance is notable, and justifies the empirical decisions made even in the absence of strong theoretical justification.

Weaknesses:
- The entire premise of the paper rests on the idea that using a subspace representation for policies is a good idea in the first place. While the empirical results of that approach reported in prior works are modestly interesting, they are far from achieving either 1) orders of magnitude in complexity reduction or 2) near zero performance reduction. Being in this 'murky middle' in the first place means this general class of approaches is unlikely to be useful in any practical setting, and raises the question of whether expanding on this research direction is of significant scientific value.
That said, since that work has passed the bar of being presented at ICLR 2022, I'll take into account that I might hold a minority view on the value of this line of research.
- The quality metric W(\alpha) of the policy is entirely computed on the replay buffer data, which is also used to update the policy parameters, which raises the question: is there a risk of overfitting to past experiences, and not actually measuring effectively the quality of the policy on new experiences? It feels like the potential for overfitting in the absence of regularization or held-out evaluation is severe (?), since the ultimate goal is to be a good anchor point for future policies.
- CSP-LINEAR is introduced but I don't see results reported for that strategy (?)


**Summary Of The Paper:**

This paper extends prior work which represents compactly a large number RL policies via a linear combination of their parameters, by making the linear combination dynamic, being able to grow incrementally as the number of policies increase and more model capacity is needed. The paper provides a heuristic to control this growth, and shows that sublinear parameter increase can be achieved at a modest cost in performance over an oracle strategy.

**Summary Of The Review:**

Nice paper, but whose usefulness is entirely predicated on the prior work (Gaya & al) being relevant, which I don't entirely agree with but am reluctant to relitigate given prior acceptance. I will rate my review as very low confidence as a result. Scientific contributions on top of that work are incremental as well.

[I had initially rated this paper a '5'. Amending to a '8' after reading other reviewer's perspective, who don't appear to share my overall sentiment, as well as thorough author rebuttals.]

---

> ### Author Response · Authors · 2022-11-11
> **Response to the review**
>
> We thank the reviewer for their constructive feedback and helpful comments. We were delighted to hear that they found our paper to be “well-written, very detailed and thorough in its evaluation” with “very good experimental analysis”, and the problem setting “sound and very relevant to lifelong learning”.
>
> &nbsp;
>
> **“Being in this 'murky middle'”**
>
> We respectfully disagree with this assessment. In fact, we’d argue that one of the biggest strengths of our approach is its flexibility, as also noted by reviewer X3Cb. When training CSP, a practitioner can control the performance / memory trade-off by setting the threshold \epsilon. For example, one can choose to maximize performance by setting no threshold (i.e. adding an anchor for each new task), or significantly reduce the memory costs by setting a high threshold. Hence, CSP contains both extremes but also everything in between. As far as we know, this type of flexibility isn’t displayed by any other popular CRL baseline. As Figure 1b shows and reviewer hkJZ also notes, “the performance of CSP is consistently better or similar to the next best benchmarks at a lower memory usage”.  Figure 2a also shows that CSP can significantly reduce the memory costs without losing much in performance, which is another advantage.
>
> &nbsp;
>
> **“Unlikely to be useful in any practical setting”**
>
> We respectfully disagree with this premise and we believe that CSP is well positioned to handle practical scenarios for the following reasons: 1) the user has control over the performance / memory trade-off as detailed above, and 2) it has the same FLOP as a single model thanks to the weight sharing subspace, which makes it fast to run at inference time.
>
> The recent surge in papers that leverage mode connectivity further indicates that this is a promising field which is of high interest for the research community [6, 7, 8].  These approaches are both theoretically grounded [4, 5] and have already shown some degree of success in supervised learning [1, 2, 3], making them good candidates to apply to other settings like CRL.
>
> Finally, these types of methods have only recently been proposed (2-3 years ago), so the field is still in its infancy [1, 2, 3]. We expect it will take a while for the community to understand how to best leverage these ideas and that it would be unwise to discourage this research so early in the process. In the end, isn’t exploration of new ideas one of the main goals of research?
>
> &nbsp;
>
> **“overfitting to past experiences”**
>
> We believe the reviewer has misunderstood how the policy selection works. After training on a new task, we sample a lot of \alpha’s and evaluate them using the Q-function and the replay buffer. Based on this, we find the best one for the extended and non-extended subspaces. Please note that this Q-function has been solely learned on one task (we reset the Q-function whenever a task changes and added it in section 3.4 to make it clear). Hence, our method shouldn’t be overfitting to past experiences.
>
> &nbsp;
>
> **“CSP-Linear not reported”**
>
> Thank you for pointing this out—this was a mistake on our part. CSP-Linear is equivalent with the “None” from Figure 2a, which is an ablation that doesn’t use a threshold so it always extends the subspace by adding one anchor for each new task. We have now removed the name of CSP-Linear since it is only used in one experiment.
>
> &nbsp;
>
> **Summary**
>
> We thank the reviewer for their valuable comments that led to further improvements to our paper. We hope to have suitably addressed your concerns with our response and paper edits. Please let us know if there is anything preventing you from further improving your support for our paper, so that we can discuss any remaining questions.
>
> &nbsp;
>
> **References**
>
> [1] Wortsman et. al.  ICML 2021. Learning Neural Network Subspaces
>
> [2] Doan et. al. 2022. Efficient Continual Learning Ensembles in Neural Network Subspaces
>
> [3] Wortsman et. al. CVPR 2022. Robust fine-tuning of zero-shot models
>
> [4] Kuditipudi et al. NeurIPS 2019. Explaining Landscape Connectivity of Low-cost Solutions for Multilayer Nets.
>
> [5] Benton et al. ICML 2022. Loss Surface Simplexes for Mode Connecting Volumes and Fast Ensembling.
>
> [6] Li et al. 2022. Branch-Train-Merge: Embarrassingly Parallel Training of Expert Language Models. 2022
>
> [7] Ilharco et al. 2022. Patching open-vocabulary models by interpolating weights.
>
> [8] ILharco et al. 2022. Editing models with task arithmetic

---

> > ### Comment · Reviewer_9jow · 2022-11-15
> > **Thank you for the reply.**
> >
> > Review updated.

---

### Official Review · Reviewer_hkJZ · 2022-10-23

**Confidence:** 4
**Correctness:** 4
**Technical Novelty And Significance:** 3
**Empirical Novelty And Significance:** 4
**Recommendation:** 8

**Clarity, Quality, Novelty And Reproducibility:**

The paper is clear and well written. All ideas are communicated effectively. The writing is high quality and the ideas are novel. The details in the Appendix should make it clear how to implement this idea andI think the work should be reproducible in theory.

**Strength And Weaknesses:**

Overall I think this is a good paper. The ideas are well presented, the experiments and ablations are reasonably convincing and the ablations are useful and interesting. I particularly liked the critic analysis of Figure 3 which, although only on a single task, does seem to indicate that the critic is learning quite well when conditioned on alpha. The paper also acknowledges inspiration from existing work and the related work section is, as far as I can tell, exhaustive.

To be honest, when I first read the idea I was surprised Q-learning with the added weights actually worked. The critic here Q(s,a, \alpha) has to estimate the value of a policy that is parameterised by some weight-vector, one component of which is also changing. However the experimental results seem quite convincing that this works in practice - the performance of CSP is consistently better or similar to the next best benchmarks at a lower memory usage.

Although I enjoyed the paper generally there are some aspects in which it could be improved which I’ve listed below.

First - I think the paper could do a better job in communicating certain relevant details. For instance the value of \epsilon typically used for training was only mentioned in the Appendix I think and is quite pertinent when looking at the results.

Similarly the Q function update actually used to train W(\alpha) should be mentioned somewhere since it is technically different from the one used in SAC. In fact, in general the explanation in Section 3.3 (`Grow the Subspace`) could be improved to reflect how \alpha is being used. On first reading, it was not clear to me that \alpha was being resampled multiple times for the same task. This was made clear by looking at the Algorithm box but it could be made more explicit in the text.

Finally, as a minor point - there is a typo in Appendix B.1 - the text says ‘two of its continuous control’ but it should be ‘three’ I think.


**Summary Of The Paper:**

This paper presents a method for continual Reinforcement Learning (learning on a sequence of related tasks) using a continuous subspace of policies. The proposed approach is quite inspired by the work of Gaya et al., 2021 in that solutions to new tasks, where possible, are parametrically represented as a convex combination of previous task parameters. These previous parameters are thought of as anchors in a subspace of policies.

When presented with a new task, the proposed algorithm attempts to learn new parameters to solve it. The subspace of policies is either expanded to include this new set or, alternatively, the task solution might be represented as a convex combination of the current subspace depending on whether the gain in performance from expanding the subspace crosses a predefined threshold \epsilon (which seems to be about 10%). This choice of expansion v/s pruning is implemented by a clever trick using a state-action value function conditioned on convex parameter sets (Q(s, a, \alpha)). This removes the need for  environment interactions to choose the weights \alpha that specify how the existing subspace should be combined as parameters for the new task.

Results are presented on locomotion and manipulation domains containing 35 different tasks in all and the proposed method usually performs the best across all domains considered.  A set of ablations on the tradeoff parameter \epsilon, scalability, learning efficiency and estimation of the alpha-conditioned critic are also presented.


**Summary Of The Review:**

Overall I think the paper is well written, the ideas are interesting and well tested on a number of domains with interesting ablations and analysis and I would recommend the paper be accepted at this venue.

---

> ### Author Response · Authors · 2022-11-11
> **Response to the review**
>
> We thank the reviewer for their positive feedback and valuable suggestions. We were delighted to hear from you that “the writing is high quality and the ideas are novel”, “the ideas are interesting and well tested on a number of domains with interesting ablations and analysis”, and “the performance of CSP is consistently better or similar to the next best benchmarks at a lower memory usage”.
>
> &nbsp;
>
> $\epsilon$ **used for training**
>
> Thank you for pointing this out — we added a mention of the threshold in Section 5.3.
>
> &nbsp;
>
> **Q-function update and** $\alpha$ **sampling**
>
> Thank you for the feedback — we added more details about the algorithm in Section 3.3 and 3.4
>
> &nbsp;
>
> **typo**
>
> Thank you –- we’ve fixed it.
>
> &nbsp;
>
> **Reproducibility**
>
> We have now uploaded our code as supplementary material.
>
> &nbsp;
>
> **Summary**
>
> We thank the reviewer for their helpful comments that further improved our paper. We hope to have suitably addressed your concerns with our response and paper edits, but don’t hesitate to let us know if you have any remaining questions.

---

### Official Review · Reviewer_X3Cb · 2022-10-24

**Confidence:** 4
**Correctness:** 4
**Technical Novelty And Significance:** 3
**Empirical Novelty And Significance:** 3
**Recommendation:** 8

**Clarity, Quality, Novelty And Reproducibility:**

The paper is well written overall, although I failed to understand some details in some cases. The presented method is new and hopefully brings long-term benefits to CRL.

**Strength And Weaknesses:**

Strengths:
- novel and appealing method
- overall quality of the paper

Weaknesses:
- in some cases, the performance of method is very close to PackNet


**Summary Of The Paper:**

The paper introduces a method of constructing a space of policies that is beneficial for future tasks.


**Summary Of The Review:**

The paper presents the method that maintains a growing set of neural networks that span a space of useful policies. Importantly the set is grown adaptively only if a new network would bring significant benefits. In this way, a sublinear number can cover all the tasks.

The major benefit, in my view, of the method is its flexibility, it can adjust the size depending on the 'capacity demand'. A concern is that the empirical results, although nice, do not clearly indicate that the presented method has clear advantage.

I have a number of questions:
- Can you explain how the training is stable when $\alpha$ changes in line 3. It'd imagine that each $\alpha$ has a much different optimal $\theta_{j+1}$ and thus some kind of jittering arises.
- In line 9 of Alg 1 is it $W_\phi$ which is trained of $Q$?
- How $Q$ function is handled? Is it reset every task? Do you keep one $Q$ for all $\alpha$?
- Do you have any expectations (or experiments) of how PackNet would perform if given $2x$ bigger network for the Brax scenarios? Namely, at the moment, PackNet uses $2x$ parameters, while CSP is $5.3$.
- It'd be nice to see a histogram of the number of used networks.
- What is the comparison of the number of parameters used in the CW10 experiments? Say, how many parameters are used for CSP and PackNet.
- I'd be great to see how the method scales with more tasks. I know it is costly, but at least one longer sequence would be great.
- As far as I understand, a very similar method could be applied in supervised continual. Could you please comment shortly on what you expect?

---

> ### Author Response · Authors · 2022-11-11
> **Answer to questions**
>
> > Can you explain how the training is stable when α changes in line 3. It'd imagine that each α has a much different optimal  $\theta_{j+1}$ and thus some kind of jittering arises.
>
> We haven’t noticed any instabilities of CSP in practice. We believe that since our algorithm is off-policy, it is more stable and can better estimate Q(s, a, $\alpha$) for the entire space. At the beginning of the project, we implemented the subspace based on PPO rather than SAC, which was more unstable and more challenging to train. This further supports our hypothesis that off-policy algorithms work better for training subspaces of policies.
>
> &nbsp;
>
> > In line 9 of Alg 1 is it $W_\phi$ which is trained of Q?
>
> $W_\phi  = \mathbb{E}_{s, a \sim B} Q_\phi(s, a, \alpha)$. In practice, we actually update the Q-function $Q_\phi(s, a, \alpha)$ using a typical SAC update. Note that given the $Q_\phi$, we can derive $W_\phi$ following Equation 2 by averaging over (s, a) pairs from the replay buffer. For simplicity, we use Q instead of $Q_\phi$ throughout the paper. More details about the algorithm can be found in Appendix C.
>
> &nbsp;
>
> > How Q function is handled? Is it reset every task? Do you keep one Q for all α?
>
> Yes, the Q-function is reset every time there is a new task and we learn one Q-function for all $\alpha$’s in the current subspace. The Q-function takes as input triples of (s, a, $\alpha$) and estimates the expected return of the policy defined by $\alpha$ on the current task (from state s, after taking action a). More details about this can be found in Appendices C.2 and C.3. We also added more clarifications regarding the Q-function in Section 3.4.
>
> &nbsp;
>
> > Do you have any expectations (or experiments) of how PackNet would perform if given 2x bigger network for the Brax scenarios? Namely, at the moment, PackNet uses 2x parameters, while CSP is 5.3
>
> Thank you for the suggestion. Following your proposal, we ran additional experiments by doubling the hidden size of PackNet (from 256 to 512) on HalfCheetah Scenarios. We noticed a slight increase of 7% in performance (from 0.85 to 0.91), but our method still outperforms it (1.23). We added the detailed results in Table 8 page 23.
> “It'd be nice to see a histogram of the number of used networks.”
> Thank you for the suggestion. We added that in Appendix D.1., Figure 7.
>
> &nbsp;
>
> > What is the comparison of the number of parameters used in the CW10 experiments? Say, how many parameters are used for CSP and PackNet.
>
> In this experiment, the growing part only concerns the number of heads (linear for PackNet, i.e. 10 which corresponds to 2,560 parameters, and almost twice less for CSP which corresponds to 1,356 parameters in average). That being said, the total number of parameters is 212,520 for PackNet and 207,380 for CSP. We believe this would make the difference in longer sequences.
> Note that PackNet uses almost double the number of parameters in the task-specific part of the model than CSP (see Table 2). In addition, CSP’s FLOP is constant (equal to that of a single network), while PackNet’s is exponential.
>
> &nbsp;
>
> > It'd be great to see how the method scales with more tasks. I know it is costly, but at least one longer sequence would be great.
>
> Sure, we are currently running experiments with 20 tasks and will update the paper with the results as soon as they finish. Note that Figure 2b already shows some results with 12 tasks, demonstrating that our method’s performance is as good as that of FT-N while its model size is only a fraction of FT-N’s (about a third). As a reminder, FT-N is our strongest baseline on this environment (HalfCheetah, see Table 1).
>
> &nbsp;
>
>  > As far as I understand, a very similar method could be applied in supervised continual. Could you please comment shortly on what you expect?
>
> Indeed, we believe something similar could be used in continual supervised learning. To adapt CSP to this setting, one could train an equivalent “value function” $W(\alpha)$ that estimates the accuracy of all models in the subspace on the given task, where each model is defined as a convex combination \alpha of the anchor parameters. $W(\alpha) = \mathbb{E}_{(x, y) \sim D} Q(x, y, \alpha)$ is computed by sampling (x, y) pairs from the corresponding dataset D and evaluating the accuracy of the \alpha model’s prediction \hat{y}. This is similar to Equation 2 in our paper where instead of sampling (s, a) pairs from the agent’s buffer B, one can sample (x, y) pairs from the corresponding dataset D. We expect the results to be broadly similar but of course this should be verified.
>
> &nbsp;
>
> **Summary**
>
> We thank the reviewer for their insightful questions and comments that led to further improvements to our paper. We hope to have suitably addressed your concerns with our response and paper edits, but don’t hesitate to let us know if you have any remaining questions.

---

> ### Author Response · Authors · 2022-11-11
> **Response to the comments**
>
> We thank the reviewer for their positive feedback and helpful suggestions. We were delighted you listed as strengths the “overall quality of the paper”, as well as “novel and appealing method” with the major benefit being “its flexibility”. We split this review in two (comments / answer to questions).
>
> &nbsp;
>
> **Performance of CSP close to PackNet**
>
> While PackNet performs quite well on Continual World (CW), its performance is much worse than that of CSP particularly on Humanoid (our most challenging Brax environment), but also on HalfCheetah and even on Ant to some degree. Note that in our experiments we use a threshold \epsilon=0.1 which achieves a good balance between performance and model size. However, one can imagine adding an anchor for each new task, which could further increase CSP’s performance.
>
> In addition, PackNet suffers from a major limitation, namely that it requires prior knowledge of the total number of tasks in order to allocate resources. If this information is not correctly specified, PackNet is likely to fail due to either not being expressive enough to handle many tasks or being too inefficient while learning only a few tasks [1]. This makes PackNet unfeasible for real-world applications where agents can face task sequences of varying lengths (including effectively infinite ones). In contrast, CSP grows adaptively depending on the sequence of tasks, so it can handle both short and long sequences without any modification to the algorithm.
>
> Moreover, we expect PackNet to struggle in adversarial sequences where the agent has to “unlearn” certain behaviors (e.g. action reversal) since many of its parameters are frozen so it lacks plasticity. In contrast, CSP should handle such settings better since it retrains the subspace for each task so the model could just move away from previous anchors. In fact, our results show that CSP outperforms PackNet on the distraction scenario where actions are inverted from a task to another (see table 8).
>
> &nbsp;
>
> **References**
>
> [1] Wolczyk et al. NeurIPS 2021. Continual World: A robotic benchmark for continual reinforcement learning.

---

> > ### Comment · Reviewer_X3Cb · 2022-11-24
> > **thank you**
> >
> > I thank the authors, and keep my recommendation to accept this paper.

---

### Official Review · Reviewer_dMX5 · 2022-10-25

**Confidence:** 3
**Correctness:** 4
**Technical Novelty And Significance:** 2
**Empirical Novelty And Significance:** 2
**Recommendation:** 6

**Clarity, Quality, Novelty And Reproducibility:**

Most of this was discussed in strengths / weaknesses.

Clarity:
- The paper is well written and explains the method well.

Quality: The paper is supports its claims well, with experiments and detailed discussion of the method.

Novelty: Discussed as a weakness above.

Reproducibility [updated]. The authors have made the code available so I believe it should be reproducible (but I didn't personally verify).

**Strength And Weaknesses:**

Strengths:
- The paper is well written and explains the method well.

- The empirical results are showing a convincing improvement, particularly over classic approaches like elastic weight consolidation.

- It's a conceptual simple approach that can probably be adapted to other algorithms (i.e. the specifics of how you learn the policy are not crucial to the method).

Weaknesses:
- The novelty is not clear. In particular, Gaya et al. (2021) [appropriately cited in the paper] have previously demonstrated the idea of using anchor points to define a subspace in parameter space for rapid online adaption. Although that is distinct from continual learning, its not a big conceptual leap.

- The CLEAR baseline is not included because "storing data from all prior tasks ... is unfeasible due to prohibitive memory costs." It would be helpful to include this baseline as a comparison (even though it is "unfair") and also to quantify the actual amount of memory necessary for these tasks. In practice, large memories are surprisingly cheap, and one could also consider store to slower access hard drives as well. I suspect store everything and train a large neural network is a surprisingly strong and practical baseline.

Minor:
  The notation for the replay buffer (Algorithm 1) is quite unusual $\mathcal{B}uf$ and looks like multiple terms. I would suggest just using $\mathcal{B}$.


**Summary Of The Paper:**

This work proposes an approach to continual reinforcement learning, where agents need to acquire a variety of skills rather than just solve a single task. The approach they propose is to learn a convex space in policy parameter space (with a mechanism to extend the subspace with a new anchor if necessary to achieve good performance on a task). In this way the number of anchors grows sublinearly with the numbers of tasks. Then, policies within this subspace can be defined by a small weight vector indicating how to combine the anchor points for the task. They test this approach on a locomotion and manipulation continual learning benchmarks where it performs well, particularly considering the resource costs.

**Summary Of The Review:**

This work adapts the idea of using anchor points to define a space space in the parameter space of policies from previously used for online adaptation to the task of continual learning, including a mechanism for adding new anchor points when it results in a large enough improvement. It's well written and supports its claim. However, it is not that novel, and conceptually only a small adaptation from earlier work. Additionally, the code is not currently available. For this reason, I'm ambivalent about acceptance.

---

> ### Author Response · Authors · 2022-11-11
> **Response to the review**
>
> We thank the reviewer for their helpful comments. We were glad they found the paper “well-written and supports its claims” and the empirical results “showing a convincing improvement”.
>
> &nbsp;
>
> **Novelty**
>
> We respectfully disagree with the reviewer’s assessment regarding the novelty of our contribution. While our method is certainly inspired by [1], we believe the differences between the two methods are substantial, as explained below. Please also note that both reviewers X3Cb and hkJZ classify our method and ideas as novel.
>
> First of all, our paper is the first to demonstrate a successful application of a subspace of policies to the continual RL setting, which is in itself a novel contribution. This is a non-trivial extension of [1] in the sequential setting. Our paper addresses the question of how to build a subspace sequentially, and how to do so efficiently. In contrast, [1] uses a fixed number of anchors irrespective of how different the test task is from the training one, which makes online adaptation less flexible.
>
> Second, the two methods differ significantly in their approach for finding a good policy in the subspace for a given task (on which the subspace was trained). [1] randomly samples k policies from the subspace, evaluates them on the task via multiple rollouts in the environment, and selects the best one. In contrast, CSP uses the learned Q-function to automatically evaluate k policies without requiring any additional rollouts in the environment.
>
> Finally, while prior work has only shown robustness to minor task variations [1], our work also demonstrates that the subspace can capture a diverse set of behaviors (see Figure 3a). More specifically, the two methods differ in the diversity of the learned subspace. [1] enforces parameter diversity which is not always correlated with functional diversity, so the subspace may still contain very similar behaviors. In contrast, the diversity of our subspace is grounded in the task sequence, so the more diverse the tasks the more diverse the policies in the subspace. Figure 3b illustrates the diversity of our learned subspace and shows that it can contain behaviors composed of previously learned skills.
>
> &nbsp;
>
>  **CLEAR baseline**
>
> We would like to kindly remind the reviewer that we are already comparing our approach with 8 relevant baselines and ablations. Nevertheless, we will add results with CLEAR for the final version of the paper. Given that the experiments can be quite expensive and we need to do a thorough HP search in order to ensure a fair comparison, we cannot yet promise we will be able to complete this by the end of the rebuttal period.
>
> We would like to highlight again the limitations of CLEAR relative to CSP. First, CLEAR uses a replay buffer with experience from all prior tasks. However, replay-based methods tend to scale poorly to long task sequences since they require an increasing amount of replay [4]. In addition, certain privacy-sensitive applications like healthcare or finance do not allow storage of personal data, so this approach is impractical in such domains.
>
> Finally, CLEAR uses a single network with constant capacity, so its plasticity decreases throughout training, resulting in poor transfer to new tasks [3]. Simply put, a single network may not be expressive enough to capture a large number of diverse behaviors, especially if there is negative transfer between the tasks. In contrast, our method CSP is more flexible, allowing the subspace (and thus the number of parameters and expressivity of the model) to grow whenever a task cannot be solved using policies from the current subspace (which is usually the case when the new task is very different from previously seen ones).
>
> &nbsp;
>
>  **Notation**
>
> Thank you for pointing this out — we changed the notation according to your suggestion.
>
> &nbsp;
>
> **Reproducibility**
>
> Apologies for the delay, we wanted to clean up the code before uploading it. We have now added the code in the supplementary material, along with instructions running experiments in the `README.md` file.
>
> &nbsp;
>
> **Summary**
>
> We thank the reviewer again for their valuable feedback that has helped us further improve our paper. We hope our response and paper updates have adequately addressed your concerns and that you will consider increasing your support for our paper. If you have any outstanding concerns, please don’t hesitate to let us know, so we can discuss them and understand what, if anything, stands between us and a strong recommendation for acceptance.
>
> &nbsp;
>
> **References**
>
> [1] Gaya et al. ICLR 2022. Learning a subspace of policies for online adaptation in reinforcement learning.
>
> [2] Wolczyk et al. NeurIPS 2021. Continual World: A robotic benchmark for continual reinforcement learning.
>
> [3] Lyle et al. 2022. ICLR 2022. Understanding and Preventing Capacity Loss in Reinforcement Learning.
>
> [4] Khetarpal et al. 2020. Towards Continual Reinforcement Learning: A Review and Perspectives.

---

> > ### Comment · Reviewer_dMX5 · 2022-11-14
> > **Response**
> >
> > Hi,
> >
> > I thank the authors for responding to my review substantively.
> >
> > I can't seem to edit my review right now (temporary OpenReview hiccup?, I will try again later) but
> >
> > 1. Now the source code is change I agree the paper is reproducible.
> > 2. I understand it might take some time to run the baseline, I wonder if you share a estimate of the total memory size of the buffer you will need to help understand how feasible such an approach is.
> > 3. I still have some concerns about the novelty but I appreciate the authors outline clearly the distinction with earlier work. I will raise my score to a 6.

---

> > > ### Author Response · Authors · 2022-11-18
> > > **Response**
> > >
> > > We kindly thank the reviewer  for taking time to answer and edit their review.
> > >
> > > &nbsp;
> > >
> > > > I understand it might take some time to run the baseline, I wonder if you share an estimate of the total memory size of the buffer you will need to help understand how feasible such an approach is.
> > >
> > > **Replay buffer size**
> > >
> > > For HalfCheetah, the replay buffer used by all the methods has a total maximum size of 2.7e7 float32  (1e6 interactions of 27 features). In comparison, a single policy (corresponding to a model size of 1 in our charts) is composed of 1.3e5 float32 parameters, i.e.  more than 200 times smaller than the replay buffer. Since we care about the model growth, the size increase of the replay buffer would be counted as part of the model size, making CLEAR intractable in its vanilla form. We believe that a reasonable buffer size is suitable for real-world problems (e.g. it opens the path to host a cheap AI module on the client side of a multiplayer video game, just like it is the case for graphic rendering).
> > >
> > > &nbsp;
> > >
> > > **The problem of not resetting the replay buffer from task to task**
> > >
> > > Nevertheless, one can imagine implementing CLEAR without increasing the size of the replay buffer (keeping a part of it dedicated to other tasks interactions). We will do our best to implement it if the paper is accepted for the camera-ready version. Yet, we have some concerns  about such settings: in many cases, sharing sensitive data from a task to another raises the question of privacy (e.g. when different tasks = different users).

---

> > > > ### Comment · Reviewer_dMX5 · 2022-12-02
> > > > **Clarification**
> > > >
> > > > Maybe I'm missing something, but aren't you saying that keeping all transitions takes ~ 2.7e7 float32 x 4 bytes per float32 ~ 1 gigabyte of memory. That seems a very practical method or have I misunderstood something?

---

> > > > > ### Author Response · Authors · 2022-12-05
> > > > > **Answer**
> > > > >
> > > > > Indeed, keeping transitions for one task in HalfCheetah does cost 1 gigabyte of memory, i.e. 100 times bigger than the actual policy. While it looks reasonable for HalfCheetah, it would scale poorly to more realistic problems like video games.
> > > > >
> > > > > As an example, Humanoid's interactions are composed of 395 features. Keeping all the transitions for one task takes around 15Gb of memory. In the CLEAR paper [1], there is an ablation study concerning the performance with respect to the size of the replay buffer. The smallest size is 5M interactions. **In the case of Humanoid, it would require around 75Gb of cached memory**. In comparison, we did not have to increase the parameter size to train our model on Humanoid.
> > > > >
> > > > > While memory-based methods may have strong advantages in some settings, we believe that they would not be competitive in this work where the core problem is maintaining a good tradeoff between model size and performance.
> > > > >
> > > > > [1] Rolnick et al. Experience Replay for Continual Learning, section 4.5

---

> > > > > > ### Comment · Reviewer_dMX5 · 2022-12-07
> > > > > > **Baseline**
> > > > > >
> > > > > > Even 75G is quite practical. It does seem like this is a fair baseline to include for at least some environments.

---

### Official Review · Reviewer_ferp · 2022-11-15

**Confidence:** 4
**Correctness:** 3
**Technical Novelty And Significance:** 3
**Empirical Novelty And Significance:** 3
**Recommendation:** 6

**Clarity, Quality, Novelty And Reproducibility:**

The paper is easy to follow, and clearly written. Using the mode connectivity idea for continual learning seems novel as described above. The authors also release code so it should be reproducible.


**Strength And Weaknesses:**


Strengths -

Significance of proposed Idea -
The problem considered  in the paper, that of continual learning for agents where they need to mitigate forgetting while also avoiding storing all data they see (due to memory constraints), is critical important for lifelong robot learning systems that keep learning different tasks. This paper uses ideas from mode connectivity (using a set of parameters and trying to generalize from the subspace they induce) and applies them to continual learning. This seems a promising direction for further exploration and the community is likely to benefit from it, since this hasn’t been studied before.

Thoroughness of evaluation -
The authors include multiple continual learning experiments to support their claims, and compare sufficiently to prior work. Across domains it appears that the proposed method gets best (or close to best) performance while also being more memory efficient, thus addressing the critical requirements for continual learning.

Weaknesses -

Tasks used for analysis
The paper includes quite a bit of analysis done on the halfCheetah environments (Fig 3- smoothness and critic accuracy, diversity and compositionally, Fig 2c - Learning Efficiency). The multiple tasks for this domain however only include parametric variations (eg - changing mass/gravity/friction etc, from Table 7). This is further away from the eventual setting where we would like to deploy continual learning, where the robot needs to learn over semantically different tasks. Related to this I’d be interested how much the subspace approach helps when the tasks are actually different (eg - pickup hammer, close box etc from continual world), but still share some structure/skills like reaching/picking/grasping/placing. While the paper does evaluate on continual world and report that it comes close to the current sota quantitatively, more analysis on this domain could strengthen the claims of the paper.


**Summary Of The Paper:**

This paper proposes an approach for continual learning using a subspace of policies (collection of ‘anchor’ parameters). Given a new task, the control policy either uses a linear interpolation of existing anchor parameters, or introduces a new anchor, and then linearly interpolates in the full set. This causes the number of stored parameters to grow sub-linearly with the number of tasks. The authors present results on locomotion environments (half-cheetah, ant, humanoid) with variations, and the continual world benchmark with robot-manipulation environments.


**Summary Of The Review:**

The paper proposes an interesting idea for continual learning which hasn’t previously been studied for RL problems, and provides sufficient evidence/evaluation through their experiments. The claims can be made stronger by using the continual world envs instead of the halfCheetah env for some of the analysis

---

> ### Author Response · Authors · 2022-11-18
> **Response to the review**
>
> We thank the reviewer for their helpful comments. We were glad to hear they find this problem “critically important for lifelong robot learning systems”, the direction “promising”, the evaluation “thorough”, and the idea “novel”.
>
> &nbsp;
>
> **"The multiple tasks for this domain however only include parametric variations"**
>
> We provide extensive analysis of CSP on a few selected scenarios in HalfCheetah since it is computationally impractical to run this analysis on all possible scenarios we train on. We would like to outline that some of our four presented scenarios are characterized by tasks that are radically different from each other in the same stream. Particularly, variations are different,drastically changing:
>
> - **The action space**: the distraction scenario totally invert the actions from task to task.
> - **The state space**: the transfer scenario proposes a task where half of the observations are masked.
> - **The transition probability function**: all scenarios are built such that fine-tuning a policy from task to task is worse than learning a policy from scratch. Like in [1], we used transfer tables (see Figure 5 page 19) to build scenarios that are more challenging than simple parametric variations (like you could have in robustness settings).
>
> To visualize the 2d and 3d plots of reward landscapes over these four scenarios, please have a look at the following [link](https://continual-subspace-policies-streamlit-app-gofujp.streamlit.app/Visualizing_the_Subspaces).
>
> &nbsp;
>
> **"How much does the subspace approach help in Continual World ?"**
>
> We thank the reviewer for this helpful recommendation. We agree that such a qualitative analysis would help understanding how CSP works in sequences of *semantically different tasks where the reward function drastically changes*. Hence, we computed and provided the reward landscapes on the 10 Continual World tasks (**see figure 8 page 30**). The learned subspace displays smoothness, diversity, and compositionality on CW as it did on HalfCheetah, so our conclusions do not change. This demonstrates that these properties hold on a wide range of settings. For convenience, we paste our analysis  below (**available page 29**):
>
> >We conducted a qualitative analysis to determine whether forward transfer also comes from the subspace’s shape (just like in HalfCheetah domain, see Figure 3b. To do so, we used the 3 first anchors created by CSP on the three first tasks of CW10, and computed the reward landscape of each of the 10 tasks. See Figure 8. We note that :
> >- **Smoothness**: As for Brax scenarios, the reward landscape remains smooth on all the figures.
> >- **Diversity**: Policies in the subspace specialize in some tasks that might share similarities. while `Push-Back` (d) is tackled by almost all policies of the subspaces, we see that policies on the lower left are good at tackling `Push-Wall` (b), `Faucet-Close` (b) and `Peg-Unplug-Side` (j). On the right side of the subspace, policies tackle `Window-Close` (i). On the top, policies are good at resolving `Hammer` (a) and `Handle-Press-Side` (f).
> >- **Compositionality**: the subspace show great compositionality on `Window-Close` (i) in a space around anchors instantiated on Hammer and `Faucet-Close` (some points reach 100\% success rate directly).
> >
> > In addition, while `Shelf-Place` (h) show zero transfer (it is the hardest task according to the learning curves in [1]), all others show different degrees of forward transfer. Interestingly, `Push-Back` (d) is also tackled by almost all policies of the subspaces, while it is not the easiest task according to the learning curves in [1].
>
> Please note that such an analysis is computationally expensive since Continual World is CPU-based.
>
>
> **Summary**
>
> We thank the reviewer again for their valuable feedback that has helped us further improve our paper. We hope our response and paper updates have adequately addressed your concerns and that you will consider increasing your score. If not, we would appreciate it if you could share what else stands in the way of strengthening your recommendation for our manuscript.
>
>
> &nbsp;
>
> **References**
>
> [1] Wolczyk et al. NeurIPS 2021. Continual World: A robotic benchmark for continual reinforcement learning.

---

### Author Response · Authors · 2022-11-11
**Answer to all reviewers**

We thank all the reviewers for their helpful comments. Following their suggestions, we added minor modifications to the paper highlighted in orange (mostly clarifications). Some supplementary material is also available:
* The code is available with some instructions to run the experiments.
* Following **Reviewer X3cb**’s suggestion, a version of Packnet with doubled hidden size has been evaluated on all HalfCheetah scenarios (see Tables 8 and 12, Appendix D).
* As requested by **Reviewer X3cb**, we added a histogram of CSP's model size when trained on HalfCheetah scenarios (see Figure 7, Appendix D).
* As requested by **Reviewer dMX5**, we added qualitative analysis of the learned subspace on Continual World (see Figure 8, Appendix D).
* We also added a few more details about our algorithm and experimental setup throughout the paper.


`Edit:` Following the recent review made by **Reviewer ferp**, we provided the reward landscape of all Continual World tasks (see figure 8 page 30, and page 29 for the analysis).

---

### Decision · Program_Chairs · 2023-01-20

**Decision:**

Accept: notable-top-25%

**Justification For Why Not Higher Score:**

As mentioned by [9jow], the provided results yield limited improvements in terms of memory usage (e.g. 2x reduction in memory). The limitation of not storing past data is also somewhat artificial for the environments considered [dMX5]. As such, while a promising direction, the method may have limited interest outside of the continual learning community.

**Justification For Why Not Lower Score:**

Clear and well-written paper proposing a novel, and well-motivated and effective solution to an important problem (continual RL). Importantly, the method offers an important hyper-parameter allowing the practitioner to trade-off performance vs memory. I expect this paper to have an impact in the broader continual learning community (beyond reinforcement learning).

**Metareview: Summary, Strengths And Weaknesses:**

The paper proposes Continual Subspace of Policies (CSP) as a scalable solution to continual reinforcement learning. At a high-level the method works by parameterizing policies as convex combinations of anchor parameters, and either growing the set of anchors with each new task or representing the optimal policy as a convex combination of previous anchors. The decision to grow the set of anchors or not comes down to a learnt “universal-value function” Q(s,a,w), which learns the Q-value for the policy derived from coefficients w. This allows them to derive a “policy value function”, with which they can evaluate arbitrary policies within the subspace. This in turn allows them to answer the counter-factual question of whether the optimal policy for the new task is best expressed as a convex combination of previous anchors, or by expanding the set of anchor points. The method is evaluated on several locomotion and manipulation domains, compares favorably across a broad range of baselines and a comprehensive set of ablations and analyses provide justification and a more in depth understanding of the method.

Overall, all the reviewers agree that this is a strong paper which tackles a fundamental problem in continual learning and thus will be of interest to the community. The method appears very effective, in particular on the more complex Humanoid domain. Of note, are the many interesting analyses and ablations of Fig 2 and 3, which visualizes the policy value function, the analysis of performance vs memory as a function of the threshold hyper-parameter.

I am happy to see that several issues of clarity, missing baselines (PACKNETX2), visualizations on Continual World have been included during the rebuttal, along with the full open-sourcing of their code. Thank you to the reviewers and authors for engaging constructively during the discussion.

One concern raised by [9jow] was that while the method does seem effective, we are far from the setting of “orders of magnitude in complexity reduction and/or near zero performance reduction.”. Indeed, from Table 1 reductions in model size are there but limited. Still, I do agree with the authors that one should not judge these methods prematurely, as e.g. such gains could very well arise when using this method at scale or in more realistic environments more amenable to transfer learning. I am thus very happy to support acceptance for this paper and wish to congratulate the authors on a great paper!

ps:  I believe the related works section could further benefit from a comparison to Successor Features and GPE/GPI which also deal with sets of policies and mechanisms for rapidly evaluating their convex combination.

**Note From Pc:**

if the above contains the word "oral" or "spotlight" please see: "oral" presentation means -> notable-top-5% and "spotlight" means -> notable-top-25%. As stated in our emails, we are disassociating presentation type from AC recommendations

**Summary Of Ac-Reviewer Meeting:**

N/A